



# Seasonal patterns of surface inorganic carbon system variables in the Gulf of Mexico inferred from a regional high-resolution ocean-biogeochemical model

Fabian A. Gomez[1,2], Rik Wanninkhof[2], Leticia Barbero[3,2], Sang-Ki Lee[2], and Frank J. Hernandez Jr.[4]

[1]Escuela de Ciencias del Mar, Pontificia Universidad Catolica de Valparaiso, Avenida Altamirano 1480, Valparaiso, Chile

[2]NOAA Atlantic Oceanographic and Meteorological Laboratory, 4301 Rickenbacker Causeway, Miami, FL 33149, USA

[3]Cooperative Institute for Marine and Atmospheric Studies, University of Miami, 4600 Rickenbacker Causeway, Miami, FL, 33149, USA

[4]Division of Coastal Sciences, University of Southern Mississippi, 703 East Beach Drive, Ocean Springs, MS, 39564, USA

Correspondence: Fabian A. Gomez (fabian.gomez@pucv.cl)

**Abstract.** Uncertainties in carbon chemistry variability still remain large in the Gulf of Mexico (GoM), as data gaps limit our ability to infer basin-wide patterns. Here we configure and validate a regional high-resolution ocean-biogeochemical model for the GoM to describe seasonal patterns in surface pressure of $CO_2$ ($pCO_2$), aragonite saturation state ($\Omega_{Ar}$), and air-sea $CO_2$ flux during 2005-2014. Model results indicate that seasonal changes in surface $pCO_2$ are strongly controlled by

temperature across most of the GoM basin, except in the vicinity of the Mississippi-Atchafalaya River System delta, where runoff largely controls dissolved inorganic carbon (DIC) and total alkalinity (TA) changes. Our model results also show that seasonal patterns of surface $\Omega_{Ar}$ are driven by seasonal changes in DIC and TA, and reinforced by the seasonal changes in temperature. Simulated air-sea $CO_2$ fluxes are consistent with previous observation-based estimates that show $CO_2$ uptake during winter-spring, and $CO_2$ outgassing during summer-fall. Annually, our model indicates a basin-wide mean $CO_2$ uptake

of 0.35 mol m$^{-2}$ yr$^{-1}$, and a northern GoM shelf (<200 m) uptake of 0.93 mol m$^{-2}$ yr$^{-1}$. The observation and model-derived patterns of surface $pCO_2$ and $CO_2$ fluxes show good correspondence, thus contributing to improved constraints of the carbon budget in the region.

## 1 Introduction

The world ocean is absorbing approximately one third of the anthropogenic $CO_2$ released into the atmosphere due to fossil

fuel burning (Sabine et al., 2004; Gruber et al 2019), resulting in a sustained decline in seawater pH and the saturation state of calcium carbonate (Orr et al., 2005). This process, commonly known as ocean acidification, has deleterious impacts on





calcifying organisms, such as corals, coralline algae, shellfish, and shell-forming plankton (Doney, 2012). Ocean acidification is disturbing marine ecosystems worldwide (e.g. Mostofa et al., 2016), demanding urgent societal responses to address coastal ecosystem impacts. A better understanding of the past and current carbon system variability at global and

regional scales is crucial to better predict ocean and ecosystem responses to enhanced $CO_2$ levels.

Significant progress has been made in the understanding of ocean carbon dynamics in coastal waters of the United States during the last fifteen years. However, many aspects remain poorly described (e.g. Chavez et al. 2007; Wanninkhof et al., 2015; Fennel et al., 2019). Uncertainties in carbon system patterns are particularly large in the Gulf of Mexico (GoM), a low-latitude semi-enclosed basin surrounded by the southern United States and eastern Mexico coast (Fig. 1). The GoM

encompasses diverse biogeochemical regimes, from the warm and oligotrophic open GoM, strongly influenced by the Loop Current and mesoscale eddies, to wide and productive continental shelves, influenced by river runoff and wind-driven coastal currents (e.g. Dagg and Breed, 2003; Zavala-Hidalgo et al., 2006; Wang et al., 2013; Muller-Karger et al., 2015; Anglès et al., 2019). Therefore, multiple dynamics modulate the GoM carbon chemistry, which makes reducing uncertainties in these biogeochemical patterns a challenging task.

Most observational studies on carbon dynamics in the GoM have been conducted in the Louisiana-Texas shelf (e.g. Cai, 2003; Lohrenz et al., 2010; Guo et al., 2012; Cai et al., 2013; Guo et al., 2012; Huang et al., 2012; 2015; Lohrenz et al., 2018; Hu et al., 2018). In this region, the Mississippi-Atchafalaya river system (MARS) has a strong influence, delivering a significant amount of freshwater, carbon, and nutrients, the latter fueling high biological production (Green et al., 2008; Lehrter et al., 2013). Enhanced primary production during spring and summer periods increases carbon uptake near the

MARS delta, which results in decreased surface partial pressure of $CO_2$ ($pCO_2$) and increased ocean uptake of $CO_2$ (Lohrenz et al., 2010; 2018; Guo et al., 2012; Huang et al., 2015; Hu et al., 2018). Subsequent sinking and remineralization of large amounts of organic carbon over the Louisiana-Texas shelf, concurrent with strong water column stratification, results in bottom acidification during the summer (Cai et al, 2011). The variability in carbon chemistry for other GoM areas has been less examined, but an increasing number of observations from dedicated research programs (e.g., Gulf of Mexico Ecosystem

and Carbon Cycle, or GOMECC) and ship of opportunity programs (SOOP) are contributing to a reduction in the spatial and temporal data gaps. Robbins et al. (2014) derived estimates of air-sea $CO_2$ fluxes over the entire GoM, concluding that the GoM basin is a $CO_2$ sink. Recently, Robbins et al. (2018) described $pCO_2$ patterns on the west Florida shelf, indicating that this region is mainly a $CO_2$ source with significant spatial and seasonal variability.

Data gaps and observational constraints limit our ability to infer carbon patterns in the ocean. Thus, regional ocean

biogeochemical models that simulate carbon dynamics at multiple timescales, are valuable tools to characterize the carbon system variability and its underlying drivers. In the GoM, several three-dimensional modeling studies addressing carbon cycle aspects have been conducted. Xue et al. (2016) used the Fennel biogeochemical model (Fennel et al. 2008; Fennel and Wilkin, 2009) to examine $pCO_2$ and air-sea $CO_2$ fluxes during 2005-2010. They reproduced observed spatiotemporal patterns across the GoM to some degree, however some discrepancies between their model results and in situ observations

were noted. For example, their model did not reproduce the decrease in surface $pCO_2$ linked to high primary production over





the MARS mixing zone (Huang et al., 2015), and spatially averaged values of model $pCO_2$ were largely overestimated in the northern GoM during summer (by more than 100 μatm in several cases). In addition, the modeled air-sea $CO_2$ flux in the northern GoM (–0.32 mol m$^{-2}$ yr$^{-1}$) was about one third of the flux derived by Huang et al. (2015) and Lohrenz et al. (2018), while the modeled flux for the deep Gulf (–1.04 mol m$^{-2}$ yr$^{-1}$) was more than twice the flux derived by Robbins et al. (2014).

In another modeling study, Laurent et al. (2017) examined near-bottom acidification driven by coastal eutrophication. Their model reproduced patterns in surface $pCO_2$, air-sea $CO_2$ fluxes, pH, alkalinity, and DIC, but the model domain was limited to the Louisiana-Texas shelf.

Discrepancies between modeling results and observations, as well as the scarcity of biogeochemical modeling studies examining GoM-wide patterns, make additional modeling efforts necessary in order to reduce uncertainty in carbon
patterns. In the present study, we use the outputs from a 15-component ocean-biogeochemical model to characterize the seasonal variability of the inorganic carbon system variables at the ocean surface, with a focus on aragonite saturation state ($\Omega_{Ar}$), $pCO_2$, as well as air-sea $CO_2$ fluxes. This paper is structured such that we: 1) describe the ocean biogeochemical model and dataset used for the study; 2) validate the model based on observations from a coastal buoy, the GOMECC-1 cruise, and SOOP; 3) describe surface inorganic carbon system variables; 4) describe air-sea $CO_2$ fluxes in coastal and ocean
domains; and 5) discuss the main model results in the context of previous observational and modeling studies.

## 2 Model and Data

### 2.1 Model

The biogeochemical model is similar to the one described by Gomez et al. (2018) and briefly detailed below, but with an additional carbon module that simulates dissolved inorganic carbon (DIC) and total alkalinity (TA). The carbon module is
based on Laurent et al. (2017) formulations, and considers a carbon to nitrogen ratio of 6.625 to link the carbon and nitrogen cycles. DIC is consumed by phytoplankton uptake, and produced by zooplankton excretion and organic matter remineralization, and affected by air-sea $CO_2$ fluxes. Changes in model TA are estimated using an explicit conservative expression for alkalinity (Wolf-Gladrow et al., 2007). Model $CO_2$ fluxes are derived with the Wanninkhof (2014) bulk flux equation. Details of the carbon module can be found in Supplement Section 1. A description of the model's nitrogen and
silica cycle components is found in Gomez et al. (2018).

The coupled ocean circulation-biogeochemical model was implemented on the Regional Ocean Model System (ROMS; Shchepetkin and McWilliams, 2005). The model domain extends over the entire Gulf of Mexico (Fig. 1), with a horizontal resolution of ~8 km, and 37 sigma-coordinate (bathymetry-following) vertical levels. A third order upstream scheme and a fourth order Akima scheme were used for horizontal and vertical momentum, respectively. A
multidimensional positive definitive advection transport algorithm (MPDATA) was used for tracer advection. Vertical turbulence was resolved by the Mellor and Yamada 2.5-level closure scheme. Initial and open-boundary conditions were





derived from a 25 km resolution Modular Ocean Model for the Atlantic Ocean (Liu et al., 2015), which includes TOPAZ (Tracers of Ocean Phytoplankton with Allometric Zooplankton) as biogeochemical model (Dunne et al., 2010). The model was forced with surface fluxes of momentum, heat, and freshwater from the European Center for Medium Range Weather
Forecast reanalysis product (ERA-Interim; Dee et al., 2011), as well as 54 river sources of freshwater, nutrients, TA, and DIC (http://waterdata.usgs.gov/nwis/qw, last accessed September 23$^{rd}$, 2018) (Aulenbach et al., 2007; He et al., 2011; Martinez-Lopez and Zavala-Hidalgo, 2009; Munoz-Salinas and Castillo, 2015; Stet et al., 2014). Monthly TA series for the MARS were derived from observations collected at the USGS stations 7373420 and 7381600. Following Stet et al. (2014), riverine DIC concentrations were calculated from observations of pH, TA, and temperature. Observational gaps in the
Atchafalaya series were filled out using linear equations linking chemical properties at the Atchafalaya station to those at the Mississippi station (Supplement Section 2). For rivers other than the MARS, we used mean climatological DIC and TA values, as the availability of data for these rivers was insufficient to generate monthly series over the entire study period. The partial pressure of atmospheric $CO_2$ was prescribed as a continuous nonlinear function, derived from the Mauna Loa monthly $CO_2$ time series (www.esrl.noaa.gov/gmd/ccgg/trends/, last accessed August 16$^{th}$, 2018) using similar curve-fitting method
that Thoning et al. (1989) (Supplement Section 3).

      The ocean-biogeochemical model in Gomez et al. (2018) was spun-up by 40 years. In the present study, an additional 9-year spin-up for the carbon system components was completed, using the basin-model boundary conditions, ERA surface forcing, and river runoff from 1981-1983. After completing the spin-up, the model was run continuously from January 1981 to November 2014, with averaged outputs saved at a monthly frequency. DIC and TA, in conjunction with
temperature and salinity, were used to derive the full set of inorganic carbon system variables, including $pCO_2$ and $\Omega_{Ar}$. The calculations were performed using the MatLab version of the CO2SYS program for $CO_2$ System Calculations (van Heuven et al., 2011), considering the total pH scale, the carbonic acid dissociation constants of Mehrbach et al. (1973) as refitted by Dickson and Millero (1987), the boric acid dissociation constant of Dickson (1990a), and the $KSO_4$ dissociation constant of Dickson (1990b).

115       For the present study, we focused on describing seasonal patterns in surface $\Omega_{Ar}$, surface $pCO_2$, and air-sea $CO_2$ flux during 2005-2014 (i.e., the last 10 years of the model run). $\Omega_{Ar}$ represents the degree of saturation of calcium carbonate ($CaCO_3$) phase aragonite, with $\Omega_{Ar}$ values less than 1 indicating undersaturation (aragonite is thermodynamically unstable, which favors dissolution), and $\Omega_{Ar}$ values greater than 1 indicating oversaturation (seawater favors aragonite precipitation). $\Omega_{Ar}$ is defined as:

$\Omega_{Ar} = [Ca^{2+}] \, [CO_3^{2-}] \, (K'_{Ar})^{-1}$

where $[Ca^{2+}]$ is total calcium concentration, which is a function of salinity, $[CO_3^{2-}]$ is total carbonate ion concentration, which is derived from the simulated DIC and TA, and $K'_{Ar}$ is the apparent solubility product of the $CaCO_3$ phase aragonite in seawater, which increases with pressure and salinity, and decreases with temperature (Mucci, 1983; Millero, 1995). At a





given pressure, temperature and salinity changes in $\Omega_{Ar}$ mainly depend on $[CO_3^{2-}]$, which are positively related to changes in
the TA to DIC ratio (Wang et al., 2013).

## 2.2 Data

Surface measurements of mole fraction of $CO_2$ ($xCO_2$), temperature, and salinity from the Central Gulf of Mexico Ocean
Observing System (Coastal Mississippi Buoy) at 30°N and 88.6°W (Sutton et al., 2014; Fig. 1) were retrieved from the
NOAA National Center for Environmental Information (www.nodc.noaa.gov, last accessed March 4, 2019). Vertical profiles
for DIC, TA, temperature, and salinity off Tampa, Florida and Louisiana were derived from measurements collected during
the GOMECC-1 cruise; Wang et al., 2013), retrieved from NOAA-AOML (http://www.aoml.noaa.gov/ocd/gcc/GOMECC1,
last accessed: March 4, 2019). Surface $pCO_2$ data were obtained from underway measurements collected onboard research
cruises and multiple ships of opportunity, and compiled by Barbero et al. (in prep). The pCO2_GoM_2018 dataset, which
contains more than 457,000 measurements in the GoM during 2005-2014 (Supplement Fig. S5), is available as data package
at NCEL.

## 3 Model-data comparison

We used data from the Coastal Mississippi Buoy to evaluate the model's ability to reproduce coastal patterns in $xCO_2$,
temperature, and salinity in the northern GoM (Fig. 2). Simulated and observed temporal surface patterns agreed reasonably
well, especially considering that the buoy is located within a region highly impacted by river runoff, strong cross-shore
gradients, and high variability in salinity, DIC and TA. We can expect therefore that relatively small changes in river plume
location (such as those derived from Mobile Bay and the Mississippi River) can significantly impact salinity and $xCO_2$,
making the exact reproduction of observed buoy patterns challenging. The best match between simulated and observed $xCO_2$
was during 2011-2012, where $xCO_2$ ranged from about 230 ppm in spring to more than 400 ppm in fall.

The pCO2GoM_2018 dataset was used to compare climatological seasonal patterns in $pCO_2$ (Fig. 3). Overall,
simulated and observed $pCO_2$ patterns were in good agreement. In the open GoM region, there was a close match between
model and observed patterns in July-December, with a relatively small model underestimation (~10 to 20 µatm) during
February-June (Fig. 3a). In the northern GoM, the largest disagreement was observed in January-February (Fig. 3b), but this
difference is most likely due to the reduced number of observations during winter in the pCO2GoM_2018 dataset
(Supplement Fig. S6). A spatial visualization of the pCO2GoM_2018 observations and model outputs is presented for each
calendar month in Fig. S6. The main spatial features were well reproduced by the model, including the $pCO_2$ minimum near
the MARS region, and the large seasonal amplitude in the western Florida shelf.

We also compared vertical patterns in DIC, TA, temperature, and salinity derived from the model, with vertical
profiles from the GOMECC-1 cruise (Fig. 4). The model reproduced the main patterns in DIC, TA, salinity, and temperature





well, especially off Tampa. Monthly averaged model DIC and TA were underestimated in the upper 200 m off Louisiana,
with the bias ranging from around 5 to 90 µmol kg$^{-1}$ for DIC and 5 to 40 µmol kg$^{-1}$ for TA, but the observations were within
or close to the simulated variable's ranges. TA and salinity were overestimated below 400 m at both stations by around 25
µmol kg$^{-1}$ and 0.3, respectively, but this bias had a limited impact on the surface properties and fluxes examined (see
following sections). Overall, our comparisons between model outputs and observations indicated that the model faithfully
reproduced relevant inorganic carbon system features and patterns, and therefore was suitable for characterizing seasonal and
spatial patterns of pCO$_2$ and $\Omega_{Ar}$ for the 2005-2014 study period.

## 4 Surface pCO$_2$ and $\Omega_{Ar}$ seasonality

Model derived patterns for surface pCO$_2$ showed significant seasonal variability across the GoM (Fig. 5). Minimum and
maximum pCO$_2$ values were generally observed during winter and summer seasons, respectively, although large spatial
differences were observed among the shelf regions. A notable model feature was observed in the central part of the northern
GoM near the MARS delta, where pCO$_2$ displayed low values year-round (<350 µatm), with a seasonal minimum in spring.
Other coastal regions less impacted by riverine discharge displayed much higher pCO$_2$ values during spring and summer
(Fig. 5b,c). The continental shelf with the maximum seasonally averaged pCO$_2$ was the west Florida shelf, where pCO$_2$
reached values greater than 450 µatm during the summer. Seasonality in modeled pCO$_2$ was strongly modulated by SST,
such that the annual amplitude for these two variables displayed very consistent spatial patterns (Fig. 6a,b; Supplement Fig.
S7). The greatest annual signal for pCO$_2$ and SST was within the northern GoM shelf and west Florida shelf, and the
smallest was in the Loop Current region. Monthly time series of modeled pCO$_2$ and SST were strongly correlated in all
regions except near the MARS delta (Fig. 6c).

The low pCO$_2$-SST correlation near the MARS delta can be explained by the role that river runoff and enhanced
primary production play as drivers of carbon system variability. This was evident in the variability of modeled pCO$_2$ along
the salinity gradient linked to the Mississippi river plume (Fig. 7). The simulated surface pCO$_2$ patterns during spring and
summer displayed a marked increase from mid to low salinities (Fig. 7a,d), which was also associated with an increase in
DIC (Fig. 7b,e). The minimum pCO$_2$ values were about 285 µatm in spring and 320 µatm in summer, at salinities close to 30
and 27, respectively. To identify the drivers of DIC variability along the salinity gradient, we displayed the simulated budget
terms for surface DIC as a function of salinity. These budget terms correspond to the air-sea CO$_2$ flux (Air-Sea), the
combined effect of advection and mixing (Adv+Mix), and the net community production (NCP), the latter representing the
difference between primary production and respiration (i.e. biologically driven changes in DIC). The derived patterns for
spring-summer showed model DIC losses at mid salinities mainly driven by NCP, indicative of a biologically induced
drawdown of pCO$_2$. During fall (Fig. 7g-i), as well as winter (not shown), NCP was much smaller than during spring-





summer, and DIC was mainly controlled by air-sea exchange and advection plus mixing processes. As a consequence, model
surface pCO$_2$ did not show a mid salinity minimum linked to phytoplankton uptake.

The simulated patterns for surface $\Omega_{Ar}$ (Fig. 8) revealed a significant meridional gradient from fall to spring, with
minimum values in the inner shelves from northern GoM and west Florida (2.5-3.6), and maximum values over the Loop
Current and west of the Yucatan Peninsula (3.9-4.1). During summer, the simulated surface $\Omega_{Ar}$ reached its maximum near
the MARS delta (>4.5), while relatively weak $\Omega_{Ar}$ gradients were observed across the open GoM region. Surface $\Omega_{Ar}$
generally displayed maximum values in winter and minimum in summer, though always well above the saturation threshold
of 1. This seasonal variation in surface $\Omega_{Ar}$ was strongly correlated to changes in the TA:DIC ratio and SST (Fig. 9a,b).
Although the seasonal patterns for $\Omega_{Ar}$ and pCO$_2$ displayed a similar phase (maximum in summer, minimum in winter), the
spatial variability of these two variables was opposite. This was most evident during spring-summer (Figs. 5b,c; 8b,c), when
the highest $\Omega_{Ar}$ and lowest pCO$_2$ values were co-located near the MARS delta, and the lowest $\Omega_{Ar}$ and highest pCO$_2$ values
were in the west Florida and northern-west GoM shelves. The annual amplitude of $\Omega_{Ar}$ displayed a similar pattern to the
annual amplitude of surface salinity, especially over the northern GoM, indicating a strong influence of river discharge on
$\Omega_{Ar}$ seasonality (Supplement Figs. S8 and S9). The correlation between $\Omega_{Ar}$ and salinity showed negative values over the
northern GoM and eastern part of the open GoM. This pattern was consistent with enhanced biological uptake of DIC
promoted by MARS's nutrient inputs (Fig. 9c).

To better describe the impact of SST in the simulated pCO$_2$ and $\Omega_{Ar}$ variability, we calculated average monthly
climatologies for temperature-normalized pCO$_2$ and $\Omega_{Ar}$ at 25°C (pCO$_{2\_at25}$ and $\Omega_{Ar\_at25}$, respectively), and compared them
with non-normalized patterns in five regions designated as the northern GoM, west Florida, western GoM, Yucatan
Peninsula, and open GoM (Fig. 10a-d; regions depicted in Fig. 1). Surface pCO$_{2\_at25}$ and $\Omega_{Ar\_at25}$ were calculated with the
CO2SYS program, using the simulated DIC, TA, and salinity patterns, and 25°C (which is close to the average SST over the
GoM basin). The strong influence of SST on model pCO$_2$ was evident when we compared the monthly climatologies for
pCO$_2$ and pCO$_{2\_at25}$ (Fig. 10a,b). Surface pCO$_{2\_at25}$ displayed much weaker annual variation than surface pCO$_2$, and the
timing for the seasonal maxima and minima largely differed. Indeed, surface pCO$_{2\_at25}$ peaked during January-February in
the northern GoM, during March in the west Florida and western GoM regions, and during February in the open GoM
regions, i.e. when pCO$_2$ was at or near its lowest levels. The comparison between $\Omega_{Ar}$ and $\Omega_{Ar\_at25}$ also revealed significant
temperature influence on model $\Omega_{Ar}$ seasonality (Fig. 10c,d). Specifically, SST amplified the annual variation in $\Omega_{Ar}$, while
having a relatively weak impact on the $\Omega_{Ar}$ seasonal phase. Both $\Omega_{Ar}$ and $\Omega_{Ar\_at25}$ were inversely related to pCO$_{2\_at25}$,
reflecting the variables dependency to DIC and TA ($\Omega_{Ar}$ increases with TA and decreases with DIC, while pCO$_{2\_at25}$ has the
opposite pattern).

Simulated climatological patterns for DIC and TA (Fig. 10e,f; Supplement Figs. S10 and S11) allowed us to
investigate the importance of DIC and TA as drivers of pCO$_{2\_at25}$ and $\Omega_{Ar\_at25}$ seasonality. In the open GoM, west Florida,
and western GoM regions, changes in TA were small, so the seasonal pattern in $\Omega_{Ar}$ was mainly due to DIC changes.
Maximum surface DIC values during late winter and early spring can be linked to increased uptake of atmospheric CO$_2$ (see



Section 5) and enhanced vertical mixing, promoted by surface cooling and winds. Alternatively, both DIC and TA played an important role modulating $\Omega_{Ar}$ seasonality in northern GoM and Yucatan Peninsula shelves. In the former, the annual
variation of DIC and TA was strongly modulated by river runoff, which is mostly associated with the MARS. Whether the MARS dilutes ocean DIC and TA depends on the season. Alkalinity in the Atchafalaya river was lower than the open GoM alkalinity year-round, whereas Mississippi alkalinity was lower than open GoM alkalinity during December-June and greater the rest of the year (Fig. S3a). The DIC of the Atchafalaya was smaller than open GoM DIC during December-May and greater from June to November, while Mississippi DIC was greater or equal to the open GoM DIC year-round (Fig. S3b).
We did not prescribe time-evolving DIC and TA for rivers other than the Mississippi River, but according to USGS records most of these other rivers have long-term average DIC and TA smaller than the oceanic values. Consequently, low TA values in the northern GoM during spring can be explained by a dilution effect, linked to maximum river discharge in the northern GoM during winter-spring. Low DIC values during spring-summer can be associated with high biological uptake, promoted by riverine nutrients and enhanced solar radiation, along with dilution (especially in spring) linked to high
discharge of low DIC waters delivered by major river inputs, like the Atchafalaya River and Mobile Bay. This is not for the case for the Mississippi river, which had DIC values greater than the open GoM. Along the Yucatan Peninsula, simulated surface DIC and TA patterns showed maximum values in summer and minimum values in winter. Coastal upwelling of DIC and TA-rich waters along the northern Yucatan Peninsula coast, reflected in a significant correlation between easterly (alongshore) winds and both DIC and TA (r = 0.65 and 0.60, respectively, with wind leading by 1 month; Supplement Fig.
S12a), influenced this seasonal pattern. The similar annual amplitude and phase for DIC and TA, as well as high TA values year-round, caused a relatively weak seasonal variability for $pCO_{2\_at25}$ and $\Omega_{Ar\_at25}$ on the Yucatan shelf. Still, a significant correlation between easterly winds and surface $pCO_{2\_at25}$ (r = 0.55) was found in the northern Yucatan coast, with $pCO_{2\_at25}$ usually peaking during spring (Supplement Fig. S12b).

## 5 Air-sea CO$_2$ fluxes

Seasonal changes in surface model $pCO_2$, mainly driven by SST changes (Fig 6c), determined strong seasonal variability in simulated air-sea $CO_2$ fluxes. As a consequence, the GoM becomes a $CO_2$ sink in winter-spring and a $CO_2$ source in summer-fall (Fig. 11a-d). An exception to this pattern occurred close to the MARS delta, which is predominantly a $CO_2$ sink year-round. In this region, the $pCO_2$ drop induced by phytoplankton uptake during spring-summer (Fig. 7a,d) determined maximum uptake of atmospheric $CO_2$ at mid salinities (seen in the air-sea exchange term in Fig. 7c,f). The greatest model
$CO_2$ uptake, above 7 mmol m$^{-2}$ d$^{-1}$, occurred over the northern GoM shelf during winter, as this region experiences the lowest surface $pCO_2$ values induced by the coldest winter conditions in the region (Fig. S7). The greatest model $CO_2$ outgassing, disregarding local peaks near major river mouths like the Mississippi river, was observed on the west Florida shelf (northern inner shelf in particular), southern Texas shelf (northern-west GoM), and western Yucatan Peninsula during the summer, ranging from ~2 to 3 mmol m$^{-2}$ d$^{-1}$ (Fig. 11c). Maximum SST values characterized summer conditions in these





regions (Fig. S7). The annual mean pattern showed modeled $CO_2$ uptake ranging from –4 to –1 mmol m$^{-2}$ d$^{-1}$ in the northern GoM, and from –2 to 0 mmol m$^{-2}$ d$^{-1}$ elsewhere (Fig. 11e). In addition, the pattern revealed areas where $CO_2$ outgassing occurred near the Mississippi River, Atchafalaya River, and Mobile Bay mouths, on the western Yucatan Peninsula, and nearshore over the west Florida shelf (Fig. 11e).

    The estimated monthly patterns for modeled air-sea $CO_2$ flux revealed prevailing $CO_2$ outgassing during May-

October in west Florida, western GoM, and Yucatan Peninsula, and June-October in the northern and open GoM (Fig.11f). The timing for the maximum $CO_2$ outgassing was June-July in the western GoM, August in west Florida and Yucatan, and September in the northern and open GoM. The timing for the maximum $CO_2$ uptake was January in the northern GoM, west Florida, and Yucatan Peninsula, and February in the western and open GoM. The model annual flux for the northern GoM, west Florida, western GoM, Yucatan, and open GoM are –2.56, –0.81, –0.60, 0.0, and –0.90 mmol m$^{-2}$ d$^{-1}$, respectively. For

the entire GoM basin, the simulated average annual flux and standard deviation was –0.97 and 2.78 mmol m$^{-2}$ d$^{-1}$ (–0.35 and 1.01 mol m$^{-2}$ yr$^{-1}$), respectively. Integrated across the entire model domain, the resulting flux was –7.0 Tg C yr$^{-1}$.

## 6 Discussion

### 6.1 Simulated carbon patterns

Characterization of historical carbon system patterns are needed to advance our understanding of carbon dynamics, as well

as to identify coastal ecosystem susceptibility to ocean acidification (Wanninkhof et al., 2015). Previous studies have described to some degree surface pCO$_2$ seasonality within the GoM (e.g. Lohrenz et al., 2010; 2018; Robbins et al., 2018), but less has been done to describe seasonal patterns for other inorganic carbon system variables. In the present study, we focused our analysis on the seasonal cycles of surface pCO$_2$ and $\Omega_{Ar}$, but seasonal patterns of surface DIC and TA were also reported. We used a similar model to the one configured by Gomez et al. (2018) for the GoM, with an extra carbon module

to simulate carbon dynamics, following model formulations described by Laurent et al. (2017). As shown in Section 3, the model simulated the main surface spatio-temporal patterns for the inorganic carbon system well. Compared to a previous basin-wide modeling effort (Xue et al., 2016), our model shows significantly less seasonal biases in surface pCO$_2$, with relatively minor pCO$_2$ underestimation during spring (<20 µatm). Further model refinements could be required for improving the representation of carbon system dynamics. These include incorporating additional model components and

processes, like dissolution and precipitation of calcium carbonate that will affect TA, improving the representation of land-ocean biogeochemical fluxes (e.g. prescribing time evolving TA and DIC for rivers other than the MARS), and increasing the model's horizontal resolution to resolve sub-mesoscale dynamics. Our current model configuration represents an important advance in the model capabilities for the GoM, capturing realistically dominant seasonal patterns.

    Simulated patterns in surface pCO$_2$ across the GoM show maximum values in spring-summer and minimum in

winter, with seasonally averaged values ranging from around 250 to 500 µatm. Seasonal variability in SST was the main





driver of surface $pCO_2$ seasonality across the GoM, except for the region around the MARS delta, where river runoff and biological uptake of DIC played a significant role during spring-summer. The $pCO_2$-SST correlation pattern derived from the model is consistent with previous observational studies, which suggested an increased correlation between $pCO_2$ and SST away from the Mississippi-Atchafalaya mixing zone, in waters associated with the surface layer from the open GoM (e.g.

Lohrenz et al., 2018). Simulated patterns in surface $\Omega_{Ar}$ showed maximum values in late summer and minimum in late winter, with most values ranging from 3 to 4.4 units. The meridional and cross-shore gradients for model surface $\Omega_{Ar}$ are consistent with patterns observed by Gledhill et al. (2008). Our model results also agree with observations by Guo et al. (2012), Wang et al. (2013), and Wanninkhof et al. (2015), which showed the most buffered surface waters off the MARS delta during summer. We found a strong positive correlation between the TA:DIC ratio and $\Omega_{Ar}$, which reflects the $\Omega_{Ar}$

dependency to changes in $[CO_3^{2-}]$. This is consistent with Wang et al. (2013), who reported spatial covariation of these two variables over the GoM and the eastern coast of USA. We also found a strong positive correlation between SST and $\Omega_{Ar}$, which can be linked to the impact of temperature on aragonite solubility (aragonite solubility decreases with temperature) and air-sea $CO_2$ fluxes (warm conditions favor surface DIC decrease due to $CO_2$ outgassing, which increases the TA:DIC ratio). Comparison between monthly climatologies for surface $\Omega_{Ar}$ and $\Omega_{Ar\_at25}$ reveals that $\Omega_{Ar}$ seasonality induced by

changes in the TA:DIC ratio tends to be reinforced by temperature-induced changes.

Air-sea $CO_2$ flux derived from the model output shows that the GoM is a $CO_2$ sink during winter-spring, and a $CO_2$ source during summer-fall. However, significant differences in the annual flux magnitude were observed among regions, which can be associated with distinct ocean-biogeochemical regimes. The northern GoM shelf, a river-dominated ocean margin strongly influenced by seasonal patterns in MARS runoff (McKee et al., 2004; Cai et al., 2013), is the coastal region

with the lowest surface $pCO_2$ and the largest $CO_2$ uptake from the model. This pattern is due to the substantial cooling experienced by the northern GoM shelf during winter (linked to its northernmost location), and the enhanced biological uptake promoted by river runoff near the MARS delta during spring-summer. Our results support the framework proposed by Huang et al. (2015) for the Mississippi river plume during spring-summer, which indicates *i*) high $pCO_2$ levels and $CO_2$ outgassing at low salinities (<20), linked to the low productivity, high turbidity, and $CO_2$ oversaturated waters delivered by

the Mississippi river; *ii*) minimum $pCO_2$ values and maximum atmospheric $CO_2$ uptake at mid salinities (20-33), as high phytoplankton production, induced by decreased water's turbidity and nutrient runoff, produces a drop in surface DIC, and *iii*) increased $pCO_2$ levels and air-sea $CO_2$ flux at high salinities (>33), as phytoplankton production declines offshore in the oligotrophic open GoM waters. In the west Florida and western GoM shelves, two coastal margins that are not strongly influenced by river runoff, temperature plays a dominant role as driver of $pCO_2$ and air-sea $CO_2$ flux seasonality. As a result,

the annually integrated air-sea $CO_2$ flux (per m$^2$) in these two shelves represents only 31% and 23% of the simulated carbon uptake in the northern GoM, respectively. In the Yucatan Peninsula, temperature is likewise the main driver of model surface $pCO_2$ and $CO_2$ flux seasonality. The zero flux in this region results from a less pronounced winter cooling, which determines a relatively weak carbon uptake during winter-spring. However, wind-driven upwelling also plays a role by increasing model surface $pCO_2$ during spring, especially nearshore. Although previous studies have documented the impact of coastal



upwelling on SST and surface chlorophyll in the Yucatan shelf (e.g. Zavala-Hidalgo et al., 2006), no study has addressed the associated impact on carbon chemistry, as insufficient inorganic carbon observations exist for this region. Further observational studies are required therefore to corroborate this dynamic.

## 6.2 $CO_2$ flux comparison

Table 2 shows mean $CO_2$ fluxes derived from our model, previous regional studies for the GoM, and global datasets. The
regional scale studies are Robbins et al. (2014; 2018), Huang et al. (2015), Xue et al. (2016), and Lohrenz et al. (2018). The global scale studies include Takahashi et al. (2009), Rödenbeck et al. (2013), Landschützer et al. (2016), Laruelle et al. (2014), and Bourgeois et al. (2016). Annual $CO_2$ fluxes for the GoM basin displayed a significant dispersion, ranging from –0.72 to +0.20 mol m$^{-2}$ yr$^{-1}$. However, the three regional studies providing basin-wide estimates (including ours) agree that the GoM is a carbon sink. We obtained an average value of –0.35 mol m$^{-2}$ yr$^{-1}$, which is comparable with Robbins et al.
(2014) and Xue et al. (2016) estimates. In contrast, two out of three basin fluxes derived from global gridded datasets, Takahashi et al. (2009) and Landschützer et al. (2016), suggest that the GoM is a weak $CO_2$ source. This discrepancy between regional and global studies most likely reflects inaccuracy in global datasets, due to the low density of pCO$_2$ observations in the GoM basin and coarse grid resolutions (5° latitude x 4° longitude in Takahashi et al. 2009 and 1° latitude x 1° longitude in Landschützer et al. 2016).

330        We obtained fluxes that are in reasonable agreement with observation-based fluxes for most of the sub-regions depicted in Figure 1. In the open GoM region, our mean flux (–0.33 mol m$^{-2}$ yr$^{-1}$) is about 70% of the flux derived by Robbins et al. (2014). For all four GoM shelf regions combined (west Florida, northern GoM, western GoM, and Yucatan), our estimated flux (–0.39 mol m$^{-2}$ yr$^{-1}$) is 20% above the value reported by Laruelle et al. (2014). In the northern GoM, our simulated flux (–0.93 mol m$^{-2}$ yr$^{-1}$) is remarkably similar to the reported fluxes of Huang et al. (2015) and Lohrenz et al.
(2018) (–0.95 and –1.1 mol m$^{-2}$ yr$^{-1}$, respectively). In the Yucatan Peninsula, our zero flux condition is close to the weak uptake condition derived by Robbins et al. (2014) (–0.09 mol m$^{-2}$ yr$^{-1}$). The major disagreement between our estimates and previous studies is on the west Florida and western GoM shelves. We determined that these two regions are carbon sinks (–0.30 and –0.22 mol m$^{-2}$ yr$^{-1}$, respectively), whereas observational studies by Robbins et al. (2014; 2018), as well as the modeling study by Xue et al. (2016), estimated a mean carbon outgassing condition. Some overestimation in our modeled
$CO_2$ uptake is possible, as the model surface pCO$_2$ in the open GoM tended to be underestimated during late winter and spring. However, the observational uncertainty in Robins et al. (2014; 2018) also needs to be considered. The dataset of underway pCO$_2$ measurements, used to generate the observed bulk $CO_2$ fluxes, has very limited spatial coverage over the western GoM. Also, this dataset has a reduced number of winter observations in west Florida and other GoM regions (only 8% of the GoM data were collected in December-February, less than 2% during January). A correct estimation of the winter
flux is important, as this season largely determines the sign of the annual flux. Indeed, excluding winter, our simulated spring to fall flux for west Florida is positive (+0.12 mol m$^{-2}$ yr$^{-1}$).





## 7 Summary and Conclusions

We configured a coupled ocean–biogeochemical model to examine inorganic carbon chemistry patterns in the GoM. The model was validated against observations from a coastal buoy, research cruises, and ships of opportunity, showing smaller

seasonal and regional bias for surface $pCO_2$ than previous modeling efforts in the region. We described seasonal patterns in surface $pCO_2$ and $\Omega_{Ar}$. Both variables show maximum values during late summer and minimum during winter and early spring. The seasonal cycle for $pCO_2$ is strongly controlled by temperature, while $\Omega_{Ar}$ follows changes in the TA:DIC ratio and temperature. Model results also indicated that river runoff and wind-driven circulation significantly influence coastal DIC and TA patterns in coastal regions, impacting $\Omega_{Ar}$, $pCO_2$, and air-sea $CO_2$ flux seasonality. Simulated fluxes show $CO_2$

uptake prevailing during winter-spring, and $CO_2$ outgassing during summer-fall. The integrated annual flux for the GoM basin is $-0.35$ mol m$^{-2}$ yr$^{-1}$ ($-4.2$ g C m$^{-2}$ yr$^{-1}$). The largest model $CO_2$ uptake is in the northern GoM shelf, linked to the most intense winter cooling, and significant biological uptake during spring-summer. The weakest $CO_2$ uptake is in the Yucatan Peninsula, mainly a consequence of the relatively warm conditions experienced by this region during winter-spring, and to a less degree wind-driven upwelling of DIC-rich waters. Sub-regional estimates are in general consistent or close to previous

observational studies, with the exception of the west Florida and western GoM shelves. We suggest that part of these discrepancies could be related to the still reduced spatio-temporal coverage in the underway $pCO_2$ measurement dataset over those two regions, especially during wintertime.

### Data Availability

The ocean–biogeochemical model outputs used in this study are available in the Network Common Data Form (NetCDF)

format on the NOAA-AOML server.

### Author contributions

SKL, RW, LB and FAG designed the study. FAG configured the model and performed the model simulations. RW and LB provided the validation dataset. FAG wrote the paper with contributions from all the authors.

### Competing interests

The authors declare that they have no conflict of interest.

### Acknowledgements





This manuscript was supported by the Northern Gulf Institute (NGI grant 18-NGI3-43), base funding of NOAA AOML, and the NOAA Ocean Acidification Program. NOAA's Ocean Acidification Program and NOAA's Climate Program Office provided funding and support for surface $pCO_2$ data collection. This research was carried out, in part, under the auspices of the Cooperative Institute for Marine and Atmospheric Studies (CIMAS), a Cooperative Institute of the University of Miami and the National Oceanic and Atmospheric Administration, cooperative agreement #NA10OAR4320143.

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





Table 1. Mean $CO_2$ flux derived from monthly model outputs during 2005-2014. Standard deviation is in parenthesis. Negative flux implies ocean $CO_2$ uptake, and positive flux $CO_2$ outgassing (shown in red). Shelf regions are depicted in Fig. 1.

| | GoM | Northern GoM Shelf | West Florida Shelf | Western GoM Shelf | Yucatan Shelf | Open GoM |
|---|---|---|---|---|---|---|
| | | | $mmol\ m^{-2}\ d^{-1}$ | | | |
| Jan | –4.03 (1.91) | –7.27 (3.17) | –4.74 (1.83) | –3.99 (2.42) | –2.63 (0.96) | –3.66 (0.98) |
| Feb | –4.07 (1.83) | –7.08 (2.54) | –4.12 (1.76) | –4.01 (2.39) | –2.45 (1.08) | –3.87 (1.15) |
| Mar | –3.70 (1.78) | –6.30 (2.76) | –3.38 (1.56) | –3.13 (1.83) | –1.80 (1.04) | –3.66 (1.14) |
| Apr | –2.39 (1.99) | –5.19 (3.36) | –1.54 (1.48) | –1.33 (1.71) | –0.24 (1.02) | –2.45 (1.21) |
| May | –0.35 (1.58) | –2.16 (3.21) | +0.32 (1.20) | +0.63 (1.86) | +1.05 (1.12) | –0.41 (0.80) |
| Jun | +1.13 (1.44) | +0.11 (2.80) | +1.62 (1.25) | +1.87 (1.93) | +1.79 (1.31) | +1.11 (0.91) |
| Jul | +1.50 (1.27) | +1.17 (2.65) | +1.84 (1.12) | +1.87 (1.70) | +1.97 (1.28) | +1.45 (0.80) |
| Aug | +1.77 (1.14) | +1.83 (2.37) | +2.57 (1.27) | +1.55 (1.16) | +1.99 (1.24) | +1.65 (0.70) |
| Sep | +1.92 (1.23) | +3.22 (2.17) | +2.28 (1.16) | +1.80 (1.36) | +1.79 (1.19) | +1.72 (0.85) |
| Oct | +1.04 (1.11) | +0.72 (1.68) | +1.15 (1.10) | +1.40 (0.95) | +1.21 (1.17) | +1.06 (0.94) |
| Nov | –1.37 (1.27) | –3.40 (1.88) | –2.00 (1.42) | –0.85 (0.95) | –0.76 (0.90) | –1.08 (0.77) |
| Dec | –3.07 (1.71) | –6.37 (2.40) | –3.68 (1.78) | –2.94 (1.88) | –1.91 (0.82) | –2.66 (0.86) |
| Annual | –0.97 (2.78) | –2.56 (4.52) | –0.81 (2.98) | –0.60 (3.41) | 0.00 (2.05) | –0.90 (2.37) |
| | | | $mol\ m^{-2}\ yr^{-1}$ | | | |
| Annual | –0.35 (1.01) | -0.93 (1.65) | -0.30 (1.09) | -0.22 (1.24) | 0.00 (0.75) | -0.33 (0.87) |
| | | | $g\ C\ m^{-2}\ yr^{-1}$ | | | |
| Annual | –4.2 (12.1) | –11.2 (19.8) | –3.6 (13.1) | –2.6 (14.9) | 0.0 (9.0) | –4.0 (10.4) |



Table 2. Comparison between annual air-sea $CO_2$ fluxes (mol m$^{-2}$ yr$^{-1}$) derived from our model results and previous studies in the Gulf of Mexico. Standard deviation is in parenthesis. Negative flux implies ocean $CO_2$ uptake, and positive flux $CO_2$ outgassing (shown in red). Shelf regions are depicted in Fig. 1.

| | Study type | GoM basin | Open GoM | All Shelves | Northern GoM Shelf | West Florida Shelf | Western GoM Shelf | Yucatan Shelf |
|---|---|---|---|---|---|---|---|---|
| Present Study | 1,3 | −0.35 (1.01) | −0.33 (0.87) | −0.39 (1.25) | −0.93 (1.65) | -0.30 (1.09) | -0.22 (1.24) | 0.0 (0.75) |
| Robbins et al. (2014) | 1,4 | −0.19 (0.08) | −0.48 (0.08) | | −0.44 (0.36) | +0.36 (0.11) | +0.18 (0.01) | −0.09 (0.05) |
| Robbins et al. (2018) | 1,4 | | | | | +0.32 (1.5) | | |
| Huang et al. (2015) | 1,4 | | | | −0.95 (3.7) | | | |
| Lohrenz et al. (2018) | 1,4 | | | | −1.1 (0.3) | | | |
| Xue et al. (2016) | 1,3 | −0.72 (0.54) | −1.04 (0.46) | | −0.32 (0.74) | +0.38 (0.48) | +0.34 (0.42) | −0.19 (0.35) |
| Takahashi et al. (2009) | 2,4,5 | +0.21 | | | | | | |
| Rödenbeck et al. (2013) | 2,4,5 | −0.13 | | | | | | |
| Landshützer et al. (2016) | 2,4,5 | +0.20 | | | | | | |
| Laruelle et al. (2014) | 2,4,6 | | | −0.33 (0.18) | | | | |
| Bourgeois et al (2016) | 2,3,6 | | | −0.79 (0.1) | | | | |

1: Regional study; 2: Global study; 3: Model-based; 4: Observational-based; 5: Gridded dataset; 6: Margins and Catchments Segmentation (MARCATS) dataset.


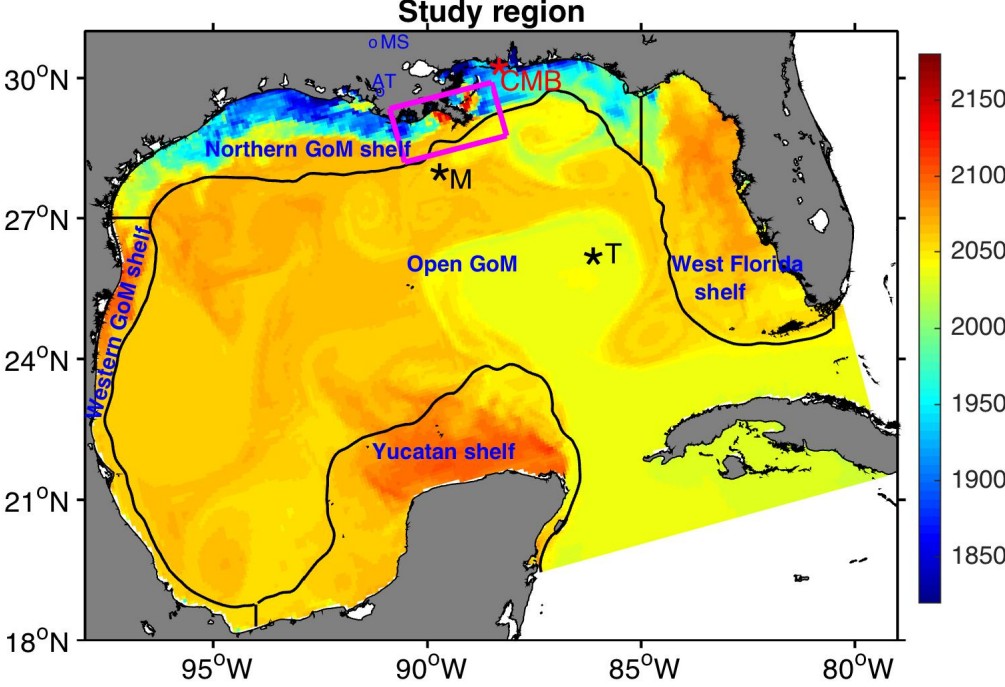

Figure 1. Model snapshot of surface dissolved inorganic carbon (mmol m$^{-3}$) during May 1$^{st}$ of 2009. Regions used to describe model results are the western GoM shelf, the northern GoM shelf, the west Florida shelf, the Yucatan shelf, and open GoM. Shelf regions are delimited offshore by the 200 m isobath. Black stars depict the location of two GOMECC stations at the Mississippi (M) and Tampa (T) lines used to validate the model. Red star depicts the location of the Coastal Mississippi Buoy (CMB). Blue circles indicate USGS stations 7373420 and 7381600 at the Mississippi (MS) and Atchafalaya (AT) rivers, respectively. Magenta polygon demarks the region near the Mississippi delta used to derive patterns in Fig. 7.





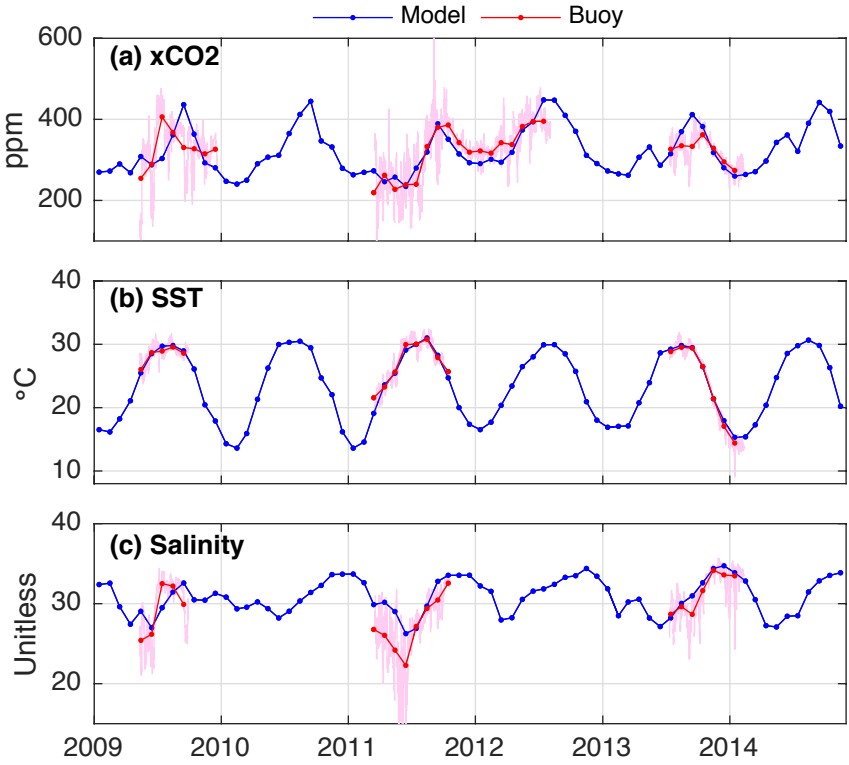

Figure 2. Time series of xCO₂, SST, and surface salinity derived from a surface mooring (Coastal Mississippi Buoy) and model outputs at 30°N and 88.6°W. Simulated and observed monthly averages are shown as blue and red lines, respectively. Buoy data (6-h interval) are depicted in magenta.



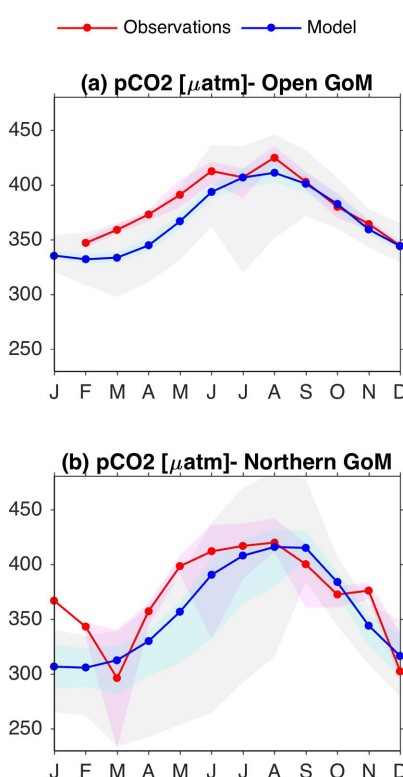


Figure 3. Observed and model seasonal patterns of pCO$_2$ over the (a) open GoM and (b) northern GoM. Light pink and cyan shades depict the observed and modeled interquartile interval, respectively. Gray shades depict the model's 5%-95% percentile interval. Observations are from Ships of Opportunity and Research Cruises conducted during 2005-2014 (ship tracks are shown in Fig. S4.1).



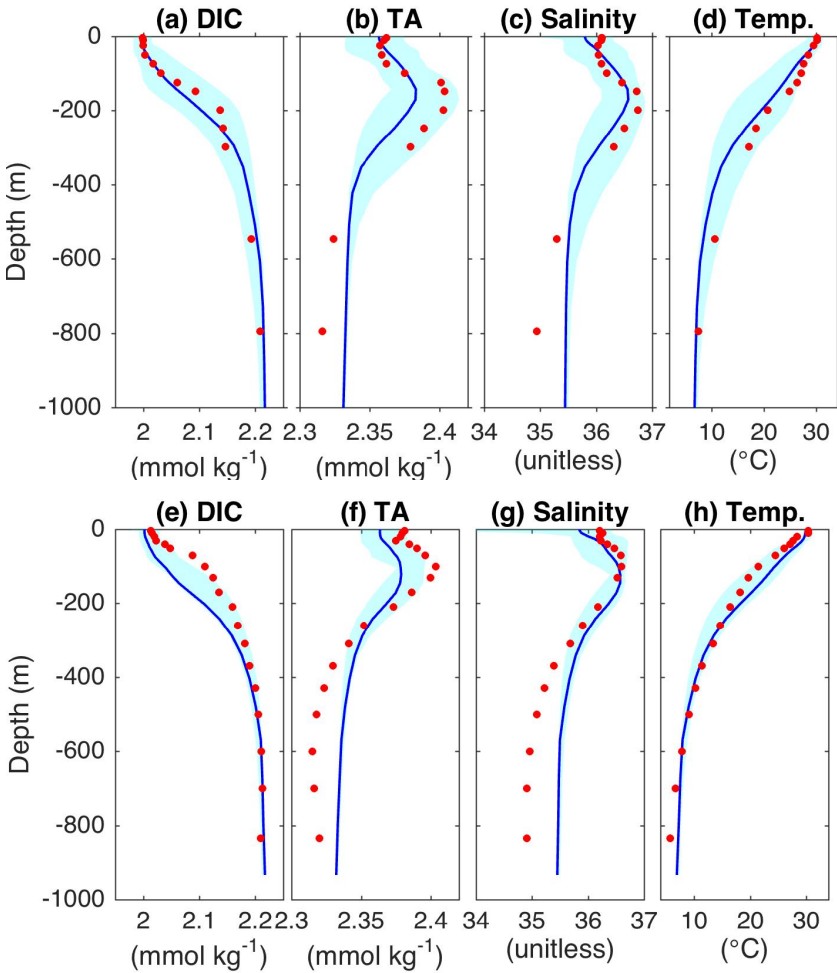

Figure 4. Comparison between profiles of (a,e) dissolved inorganic carbon (DIC), (b,f) total alkalinity (TA), (c,g) salinity, and (d,h) temperature from monthly model outputs (blue lines) and GOMECC-1 data (red dots). The model's variables range

for June-August during 2000-2014 is also shown as cyan shade. (a-d) and (e-f) show the profiles for the most oceanic station at the Mississippi and Tampa lines (see Figure 1).



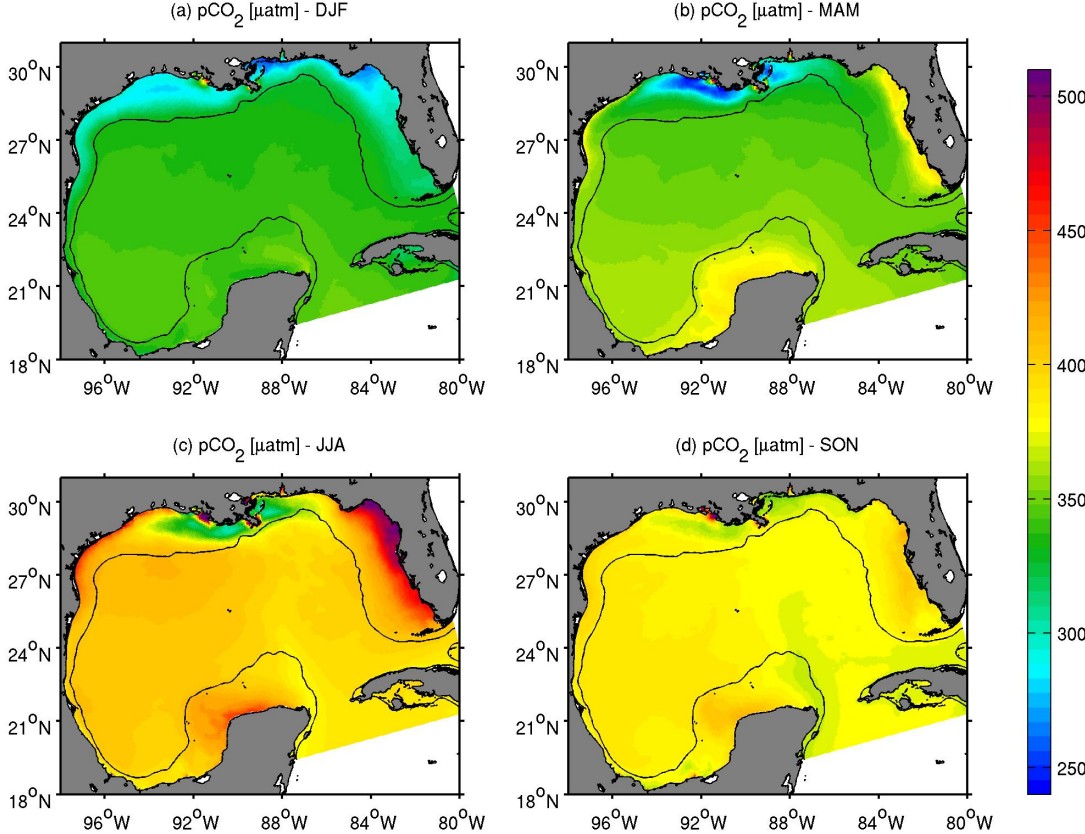

Figure 5. Mean model surface $pCO_2$ (*u*atm) in winter (DJF), spring (MAM), summer (JJA), and fall (SON) from 2005-2014.
The black contour depicts the 200 m isobath.






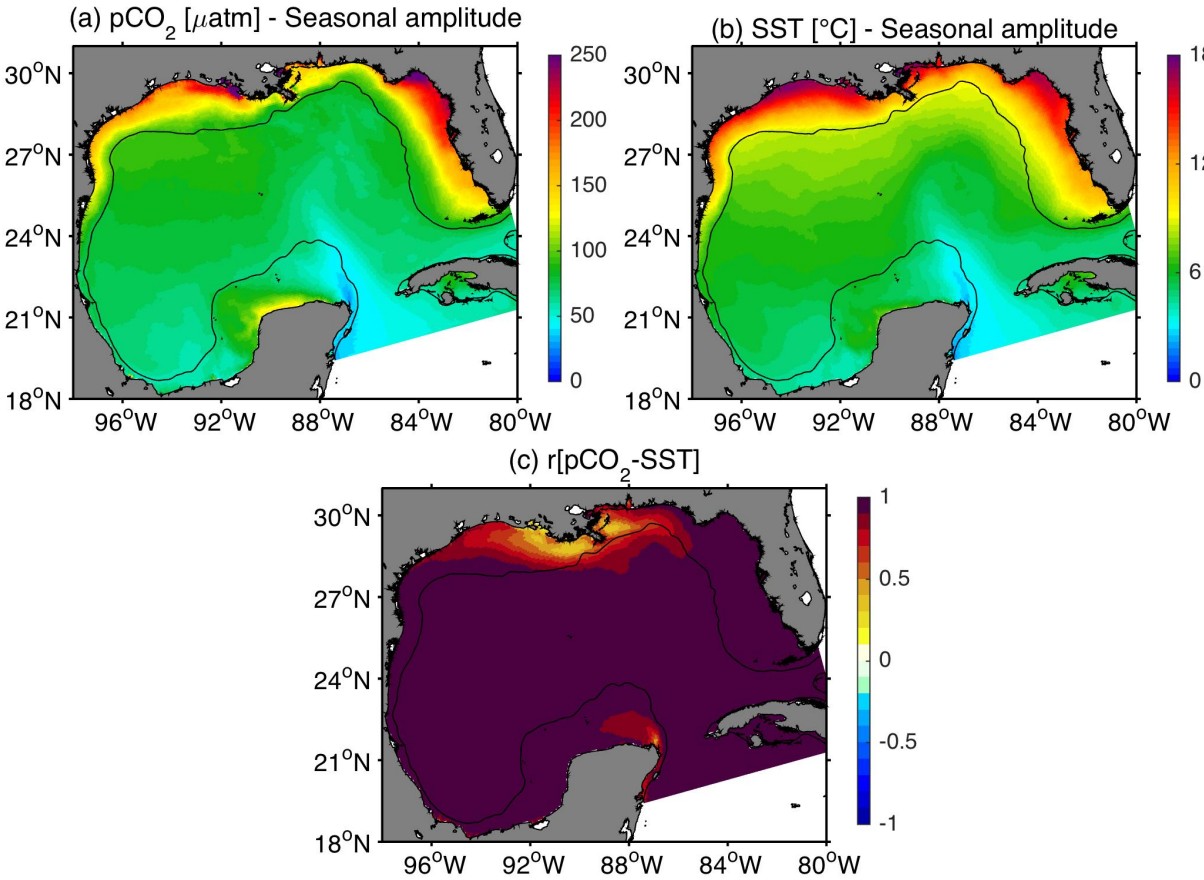

Figure 6. (a,b) Seasonal amplitude patterns for model surface pCO2 and SST. The seasonal amplitude is the difference between the maximum and minimum values from monthly climatologies at each grid point (c) Correlation between surface model $pCO_2$ and SST. Black contour depicts the 200 m isobath.




Figure 7. Mean patterns of simulated surface variables as a function of salinity near the Mississippi river (magenta polygon in Fig. 1) during spring (a-c), summer (d-f), and fall (g-i): (a,d,g) pCO$_2$ and pCO$_2$ normalized to 25°C; (b,e,h) dissolved inorganic carbon (DIC) and total alkalinity (TA); (c,f,i) budget terms for DIC: advection plus mixing (Adv+Mix), air-sea CO$_2$ flux (Air-Sea), and net community production (NCP). Thin dashed lines demarcate the interquartile interval (between
percentiles 25% and 75%). Only results for salinities greater than 17 are shown, since the spatio-temporal resolution from the monthly model outputs did not resolve features at lower salinities well.





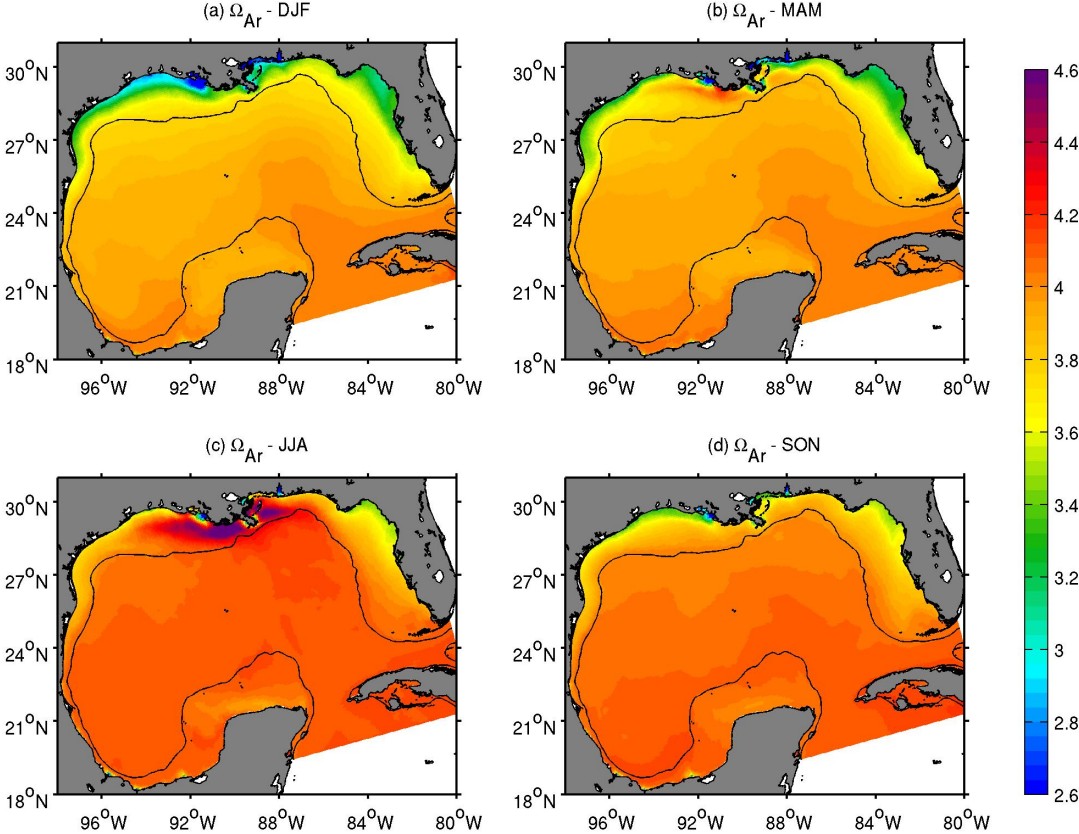

Figure 8. Mean model surface aragonite state in winter (DJF), spring (MAM), summer (JJA), and fall (SON) from 2005-2014. The black contour depicts the 200 m isobath.






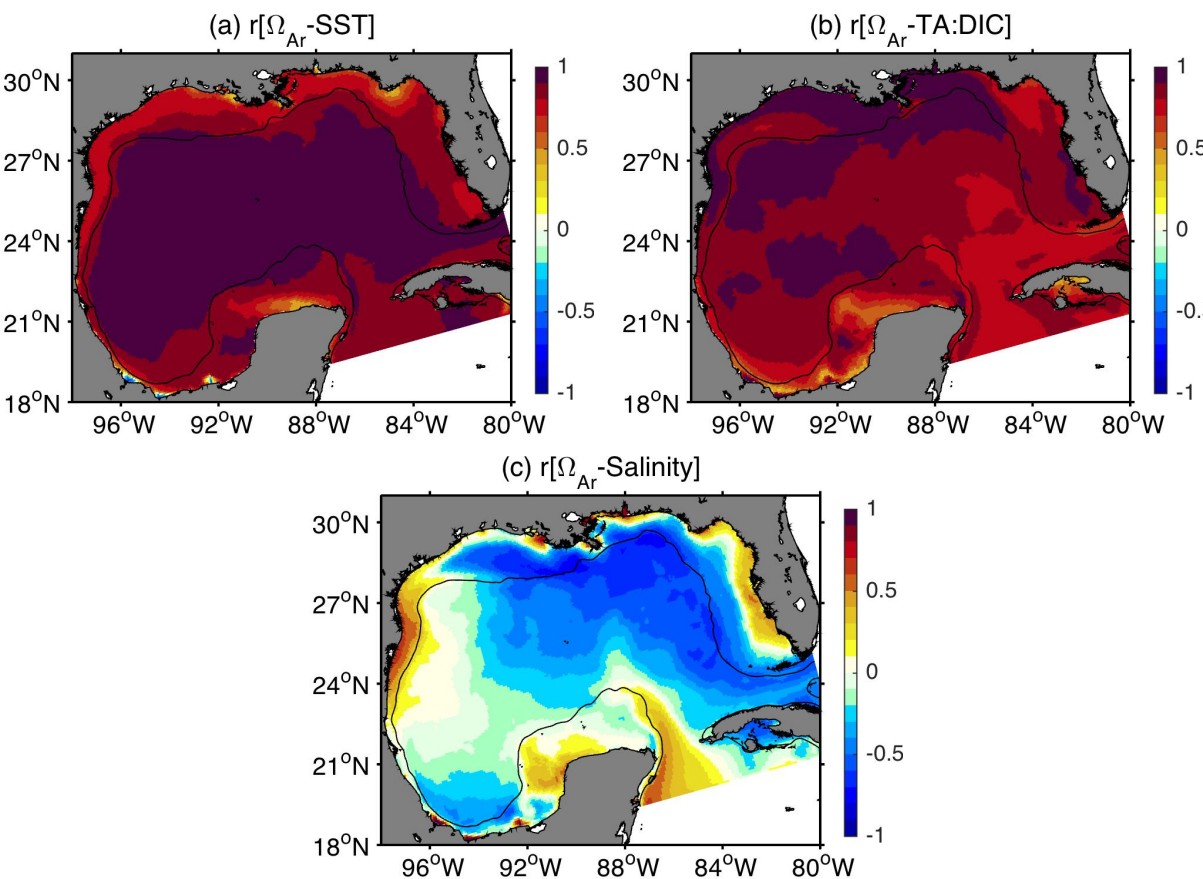

Figure 9. Correlation between surface aragonite saturation state and surface (a) temperature, (b) TA to DIC ratio, and (c)
salinity. The black contour depicts the 200 m isobath.

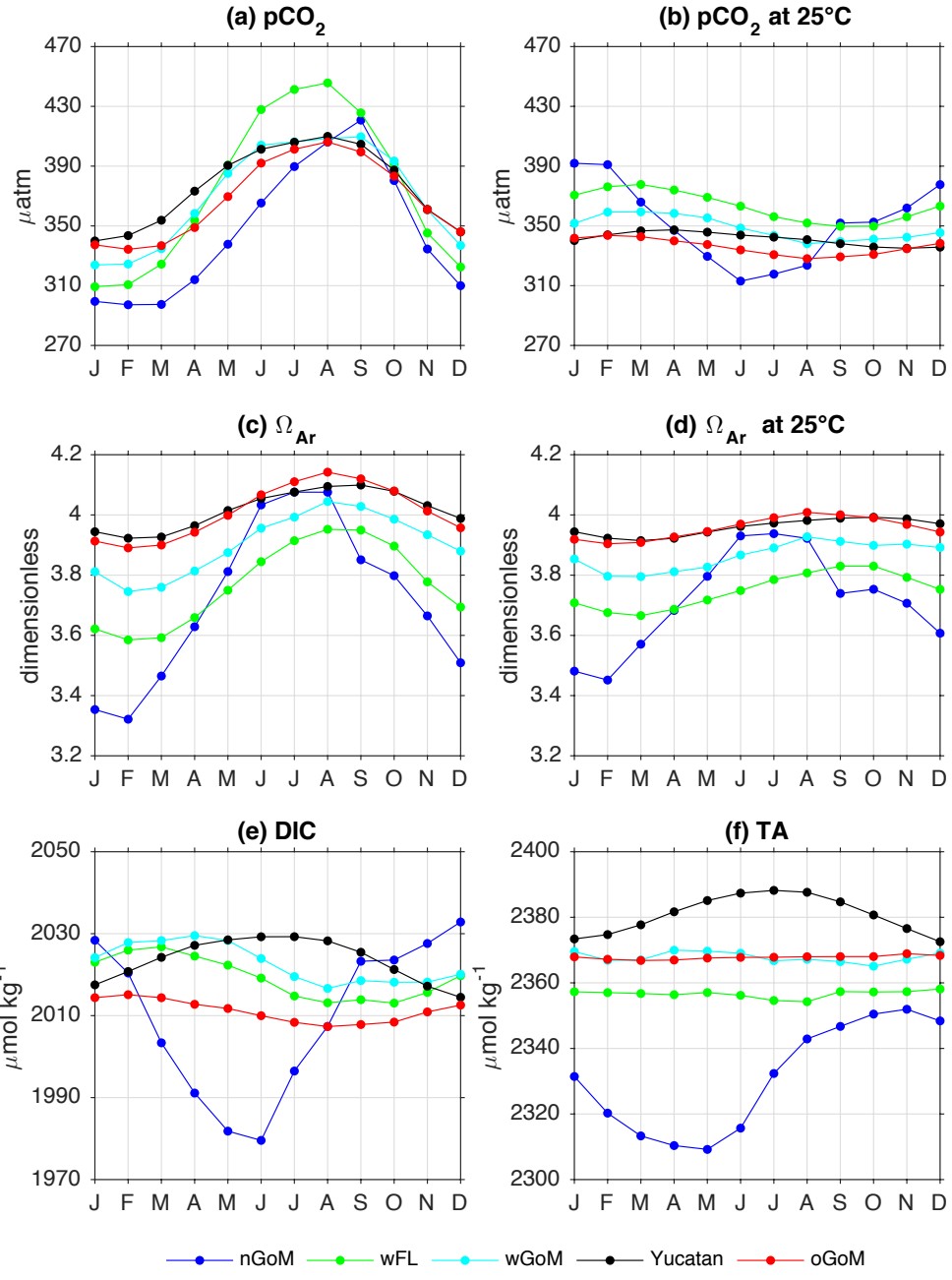

Figure 10. Monthly climatology for model (a) $pCO_2$, (b) $pCO_2$ at 25°C, (c) aragonite saturation state ($\Omega_{Ar}$), (d) $\Omega_{Ar}$ at 25°C, (e) dissolved inorganic carbon (DIC), and (f) total alkalinity (TA) in northern GoM shelf (nGoM; blue), west Florida shelf (wFL; green), western GoM shelf (wGoM; cyan), Yucatan shelf (black), and open GoM (oGoM, red). Patterns were derived for 2005-2014.




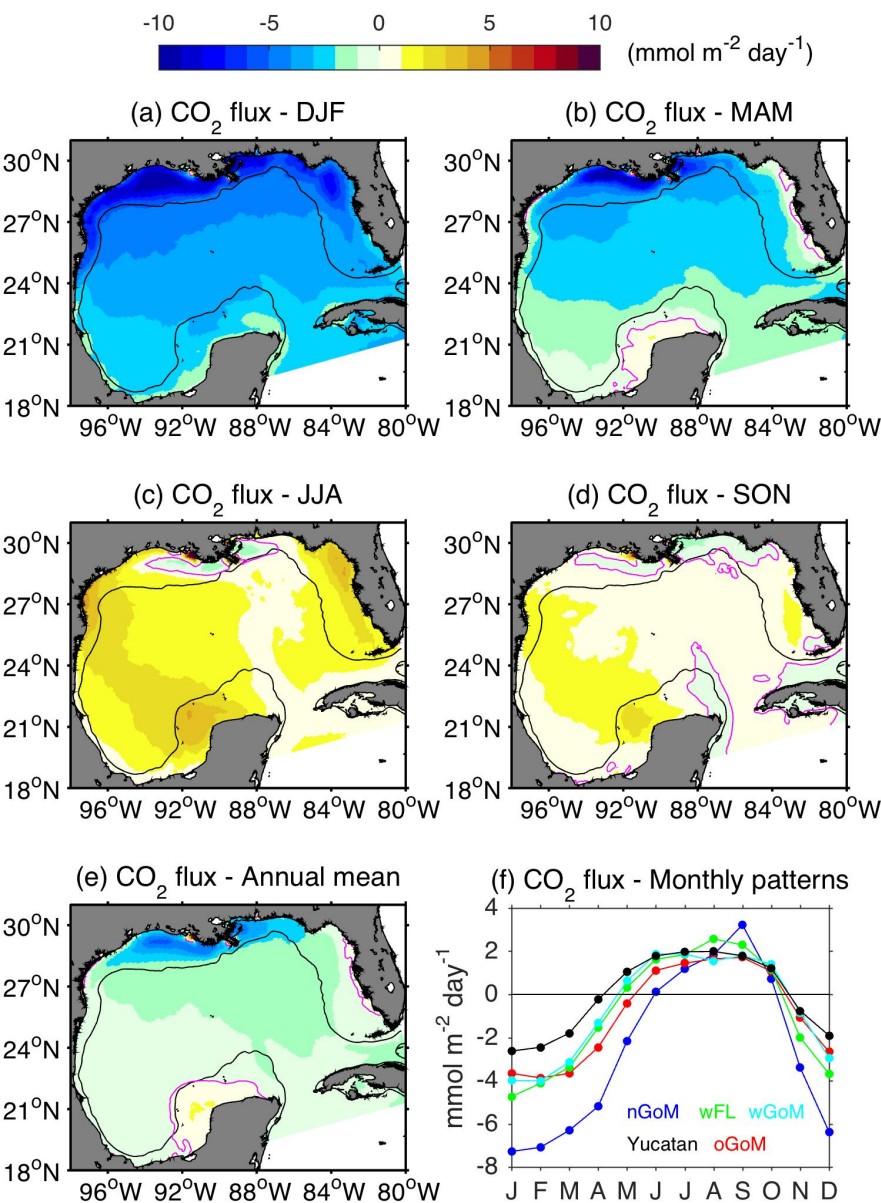

Figure 11. Model air-sea $CO_2$ fluxes (mmol m$^{-2}$ day$^{-1}$) patterns during 2005-2014. (a-d) Spatial mean patterns for (a) winter (DJF), (b) spring (MAM), (c) summer (JJA), and (d) fall (SON). (e) Spatial annual mean. (f) Monthly climatology for the northern GoM shelf (nGoM; blue), west Florida shelf (wFL; green), western GoM shelf (wGoM; cyan), Yucatan shelf (black), and open GoM (oGoM, red). Negative (positive) flux implies ocean uptake (degassing). Magenta contours in panels a-e depict 0 mmol m$^{-2}$ day$^{-1}$, and black contours the 200 m isobath.