# Peer review of "Seasonal patterns of surface inorganic carbon system variables in the Gulf of Mexico inferred from a regional high-resolution ocean-biogeochemical model"

_Biogeosciences, 2019_

## Referee Comment (RC1) · Anonymous Referee #1 · 28 Nov 2019

In their manuscript Gomez et al expand their Gulf of Mexico biogeochemical model (Gomez et al., 2016) to include carbonate chemistry. After some validation, focused on this new part of the model, the authors describe the annual climatological spatial and temporal patterns of the modeled (or derived) carbonate chemistry variables and discuss the air-sea $CO_2$ flux in their model in comparison to former global or regional studies. The manuscript is well written, the method apropriate, the figures are good quality and the conclusions supported by the results. My main general comment is to expand the discussion of your results. Currently it is not clear what is the novelty of

your estimates in air-sea CO2 flux in comparison to the previous regional work. Your discussion mostly focus on showing that your model is able to simulate the regional carbon dynamics. Could you also provide some discussion what your results mean in term of the Gulf regional carbon dynamics, i.e. for each of your regions. For instance you mention that there is a need "to identify coastal ecosystem susceptibility to ocean acidification" but this is not discussed further, i.e. with respect to your results. A few specific comments are provided below.

L106-107: is it 40/9 years of the same annual cycle? Is 9 years sufficient for the carbon system, e.g. for the deep Gulf it seems short, and how did you assess that the carbon system was spun-up?

L109: It would have been interesting to see model output for the period January 1981 to November 2014.

L144-145: a more accurate statement would be "Overall, simulated and observed pCO2 patterns agreed with observations"

L146-147: is it possible that your model generally overestimates surface primary production, resulting in a lower surface pCO2?

L144-151: There is obviously a very large difference in the shape of the observed versus modeled pCO2 time series in NGoM (Figure 3b). This should be discussed. Why is there a strong dip in pCO2 in March in the observations and why it doesn't occur in the model? January-February observations are odd. Figure 3b also shows that there is a 1 month delay in the modeled pCO2, which tend to follow more the temperature cycle. Can you discuss these discrepancies?

L152-160: in Figure 4 caption can you add location information, i.e. off Tampa (upper panel?) and off Louisiana (lower panel?) and refer to Figure 4a and Figure 4b when appropriate.

L155-156: the 0-200m difference in DIC and TA is quite large. You need to provide

more discussion here to gain confidence in the results presented below. What is the source of this discrepancy?

L230: "This is not the case..."

---

## Referee Comment (RC2) · Anonymous Referee #2 · 30 Nov 2019

General Comments

This is a very well written manuscript, with good figures, and scientific arguments that are interesting and well-constructed. The authors appear to have made a significant advance in understanding the regional and temporal variability of seawater CO2 chemistry and air-sea CO2 fluxes in the Gulf of Mexico. There is potential for expanding the discussion (see specific comments). However, one of the aspects I appreciated of the manuscript as it stands was the concise length: the authors should try to limit text additions in response to reviewers and balance with (careful) trimming. Overall, I congratulate the authors for an excellent submission and recommend only minor revisions.

Specific Comments

"air-sea flux": Most readers will read this as AIR-TO-SEA flux. Therefore it should be positive when the ocean is a sink and negative when the ocean is a source (as in e.g. Xue et al., 2016). I recommend that the authors either: a) reverse the signs on all "air-sea" fluxes, or b) speak of "sea-air" fluxes, which is an alternative convention.

L112-113: The carbonate chemistry equilibrium constants from Mehrbach et al. (1973) refitted by Dickson and Millero (1987) might not be optimal for the salinities <20 psu explored by this study (cf. Fig. 6; see Millero 2010, and stated validity ranges in the van Heuven code).

L148-150: Looking at Fig. S6 it seems that the mismatch may be primarily due to interannual variability and temporal undersampling. The observations in Fig. S6 do suggest a strong decrease in pCO2 on the northern GoM shelf during JFM, but they appear to be from single cruises and perhaps a single year (?), while the model results are averaged over 10 years. Perhaps clarify about this.

L272-273: I am left wondering exactly why, compared with the Xue et al. model, the present model apparently simulates stronger biological DIC uptake and associated pCO2 decrease in the MARS region, sufficient to turn this region into a year-round CO2 sink (cf. present Fig. 11 vs. Xue et al., 2016, Fig. 7). Is it possible that the apparent improvement in fit to surface pCO2 observations (present Fig. S6) could be for the wrong reasons? Can the stronger biological uptake be corroborated with other observations (e.g. nutrient drawdown)? Also, it seems that the large-scale seasonal variability in CO2 flux, here driven primarily by temperature, is stronger than in Xue et al. Is this true, and if so, why?

Millero, F.J. 2010. Carbonate constants for estuarine waters. Marine and Freshwater Research 61(2) 139–142

Technical Corrections

L40: ON the Louisiana-Texas shelf

L55: No comma after 'timescales'

L63,64: Flip signs on 0.32 and 1.04

L78: Delete 'and briefly detailed below,'

L83: derived USING

L84: Maybe you can save some space by removing "Supplement" (10 instances) and just referring to Section S1, Fig. S1, etc. Check journal requirements.

L88,89: Hyphenate 'third order' and 'fourth order'

L97,98: 'Stets' not 'Stet'

L98,99: I do not see a description of how to calculate river DIC from river (pH, TA, T) in the Stets et al. (2014) listed in the References. Is this the correct reference?

L106: FOR 40 years

L124: ...temperature and salinity, changes...depend on [CO3 2–] and are...

L125: TA:DIC

L134: as a data package

L147: ON the northern GoM SHELF (assuming the statistics are restricted to the region shown in Fig. 1).

L148: MAY BE due...

L156: ...ranges DURING JUNE-AUGUST 2000-2014.

L190: maximum values in SUMMER and minimum in WINTER

L202-203: Refer to regions exactly as defined in Figure 1, not omitting "shelf"

Figures

Figure 2: Assuming this is practical (not absolute) salinity, I disagree that it should be "unitless". You could neatly specify which definition is used with the unit [psu], which for me is a perfectly valid and informative dimensionless unit.

Figure 3: Add sentence to caption saying what the red/blue lines are (presumably mean values over all model grid points and observations within the regions defined in Fig. 1). Reference should be to Fig. S5 not S4.1. Refer to northern GoM SHELF (assuming that the statistics are restricted to the shelf region).

Figure 4: ON the Mississippi and Tampa lines

---

## Author Comment (AC1) · 16 Jan 2020

*We highly appreciate the reviewers for all their valuable comments and suggestions. We have included our responses in blue.*

**My main general comment is to expand the discussion of your results. Currently it is not clear what is the novelty of your estimates in air-sea CO2 flux in comparison to the previous regional work. Your discussion mostly focuses on showing that your model is able to simulate the regional carbon dynamics. Could you also provide some discussion what your results mean in term of the Gulf regional carbon dynamics, i.e. for each of your regions? For instance you mention that there is a need "to identify coastal ecosystem susceptibility to ocean acidification" but this is not discussed further, i.e. with respect to your results.**

*We agree that the discussion section could be expanded. However, as indicated by the second reviewer, we would prefer a limited extension in order to keep a concise manuscript length. The following components will be added:*

*1) Sea-air $CO_2$ fluxes:*

*a) This is the first regional modeling effort that closely agree with the observed fluxes by Huang et al. (2015) and Lohrenz et al. (2018) in the northern GoM shelf, which is the region in the GoM-basin where the carbon fluxes are better constrained.*

*b) Our results show large discrepancies with the modeling results by Xue et al. (2016). Surface $pCO_2$ observations indicate that the main reasons of these discrepancies are that Xue et al model tends to overestimate $pCO_2$ in the coastal regions and underestimated pCO2 in the open GoM, which leads to important biases in their $CO_2$ fluxes.*

*c) Increasing sea temperature due to anthropogenic climate change in the GoM (e.g. Liu et al., 2015) could significant modify $CO_2$ fluxes in the next decades, potentially turning weak uptake regions, such as west Florida and western GoM, into carbon sources.*

*2) Coastal ecosystem susceptibilities to ocean acidification:*

*a) Our model indicates minimum surface $\Omega_{Ar}$ on the northern GoM and west Florida inner shelves during winter-spring, with values ranging from 2.8 to 3.4. Instead, minimum surface $\Omega_{Ar}$ on the western GoM and Yucatan shelves tend to be greater than 3.4. Higher ecosystem resilience to surface ocean acidification could be expected therefore in these last regions.*

*b) Surface $\Omega_{Ar}$ patterns do not reflect vulnerability of benthic organisms to ocean acidification, since $\Omega_{Ar}$ values for surface and bottom layers can largely differ. This is the case for the Louisiana shelf during summer, which displays maximum surface $\Omega_{Ar}$ values (>4.2) linked to high biological uptake, but low bottom $\Omega_{Ar}$ values (<2.5; not shown) due to high remineralization and weak bottom ventilation (Cai et al., 2011).*

**Specific comments:**

**L106-107: is it 40/9 years of the same annual cycle? Is 9 years sufficient for the carbon system, e.g. for the deep Gulf it seems short, and how did you assess that the carbon system was spun-up?**

*We consider that a 9-year simulation is an appropriate time for spinning-up the model, as the simulated DIC and TA patterns in the upper ~800 m reached a periodic steady state after 3 or 4 years. This was checked by visual inspection of the model outputs, and estimating DIC and TA linear trends. Deep layer pattern could take longer, but this most likely has a limited influence in the surface properties of the GoM.*

**L109: It would have been interesting to see model output for the period January 1981 to November 2014.**

*We focused on seasonal dynamics only in this paper because interannual variability will be addressed in a following study.*

**L144-145: a more accurate statement would be "Overall, simulated and observed pCO2 patterns agreed with observations"**

*The change will be done accordingly.*

**L146-147: is it possible that your model generally overestimates surface primary production, resulting in a lower surface pCO2?**

*Although an overestimation of surface primary production could not be discarded, the comparison between simulated and observed primary production pattern did not reveal any evident bias (see Fig. 6 in Gomez et al., 2018).*

**L144-151: There is obviously a very large difference in the shape of the observed versus modeled pCO2 time series in NGoM (Figure 3b). This should be discussed. Why is there a strong dip in pCO2 in March in the observations and why it doesn't occur in the model? January-February observations are odd. Figure 3b also shows that there is a 1 month delay in the modeled pCO2, which tend to follow more the temperature cycle. Can you discuss these discrepancies?**

*The pCO2GoM_2018 dataset has very few observations during wintertime. Only 8% of the GoM data were collected in December-February, less than 2% during January. Indeed, for the northern GoM, the observations for January were derived from only one cruise. Consequently, there is a large observational uncertainty in winter (an aspect that is discussed in Section 6.2.). We will include the following sentences to clarify.*

*"In the northern GoM, the largest disagreement was observed in January-February (Fig. 3b), but this difference is most likely due to the reduced number of observations during*

*winter in the pCO2GoM_2018 dataset (Supplement Fig. S6). Indeed, January observations came from only one cruise, which largely increases observational uncertainty."*

**L152-160: in Figure 4 caption can you add location information, i.e. off Tampa (upper panel?) and off Louisiana (lower panel?) and refer to Figure 4a and Figure 4b when appropriate.**

*The change will be done accordingly.*

**L155-156: the 0-200m difference in DIC and TA is quite large. You need to provide more discussion here to gain confidence in the results presented below. What is the source of this discrepancy?**

*We recognize that there are important differences between modeled and observed profiles in the upper 200 m for the Mississippi line, although observed values are within or close to the simulated variable's range. The open GoM region off Louisiana, where the Mississippi line extends, is strongly influenced by the Mississippi river runoff, displaying a large spatiotemporal variability. Relatively minor differences between the observed and simulated cross-shore fluxes could lead to important differences between the observed and simulated vertical distribution of DIC and TA. We will include the following sentences in the new manuscript version:*

*"Monthly averaged model DIC and TA were underestimated in the upper 200 m off Louisiana, with the bias ranging from around 5 to 90 $\mu$ mol kg$^{-1}$ for DIC and 5 to 40 $\mu$ mol kg$^{-1}$ for TA, but the observations were within or close to the simulated variable's ranges. In part, these model-observation differences could be partly due to misrepresentation of cross-shore transport in a region strongly influenced by the Mississippi river runoff."*

**2. Response Reviewer 2**

**This is a very well written manuscript, with good figures, and scientific arguments that are interesting and well-constructed. The authors appear to have made a significant advance in understanding the regional and temporal variability of seawater CO2 chemistry and air-sea CO2 fluxes in the Gulf of Mexico. There is potential for expanding the discussion (see specific comments). However, one of the aspects I appreciated of the manuscript as it stands was the concise length: the authors should try to limit text additions in response to reviewers and balance with (careful) trimming. Overall, I congratulate the authors for an excellent submission and recommend only minor revisions.**

**Specific Comments**

**"air-sea flux": Most readers will read this as AIR-TO-SEA flux. Therefore it should**

**be positive when the ocean is a sink and negative when the ocean is a source (as in e.g. Xue et al., 2016). I recommend that the authors either: a) reverse the signs on all "air-sea" fluxes, or b) speak of "sea-air" fluxes, which is an alternative convention.**

*The corrected paper version will use "sea-air" flux instead of "air-sea".*

**L112-113: The carbonate chemistry equilibrium constants from Mehrbach et al. (1973) refitted by Dickson and Millero (1987) might not be optimal for the salinities <20 psu explored by this study (cf. Fig. 6; see Millero 2010, and stated validity ranges in the van Heuven code).**

*Although using Millero (2010) constant for low salinity regions (S<20) could be optimal, the deviation between Millero and Mehrbach et al. is not significant until S<5. Very little of the study area has S < 20. Indeed, due to the spatial resolution of our model (~8 km), the simulated surface salinity values were always greater than 5, and only a small fraction (0.1%) of the surface layer outputs have salinities smaller than 20. Therefore, we believe that using Mehrbach et al. constants does not significantly bias our model results. We will add this discussion in the revised paper.*

**L148-150: Looking at Fig. S6 it seems that the mismatch may be primarily due to interannual variability and temporal undersampling. The observations in Fig. S6 do suggest a strong decrease in pCO2 on the northern GoM shelf during JFM, but they appear to be from single cruises and perhaps a single year (?), while the model results are averaged over 10 years. Perhaps clarify about this.**

*In the northern GoM, the observations for January, February, and March were derived from one (2009), three (2009, 2010, 2012), and two (2009, 2010) cruises, respectively. Certainly, as mentioned in the Discussion section (L344-344), the limited number of pCO$_2$ observations collected during wintertime increases the observational uncertainty. We will include the following sentences to clarify.*

*"In the northern GoM, the largest disagreement was observed in January-February (Fig. 3b), but this difference is most likely due to the reduced number of observations during winter in the pCO2GoM_2018 dataset (Supplement Fig. S6). Indeed, January observations came from only one cruise, which largely increases observational uncertainty."*

**L272-273: I am left wondering exactly why, compared with the Xue et al. model, the present model apparently simulates stronger biological DIC uptake and associated pCO2 decrease in the MARS region, sufficient to turn this region into a year-round CO2 sink (cf. present Fig. 11 vs. Xue et al., 2016, Fig. 7). Is it possible that the apparent improvement in fit to surface pCO2 observations (present Fig. S6) could be for the wrong reasons? Can the stronger biological uptake be corroborated with other observations (e.g. nutrient drawdown)? Also, it seems that the large-scale seasonal variability in CO2 flux, here driven primarily by temperature, is stronger than in Xue et al. Is this true, and if so, why?**

*We feel confident that our model produces realistic results. A series of studies have recognized the importance of biological uptake as a main driver of carbon patterns in the northern GoM (e.g. Guo et al. (2012) and Huang et al. (2015)). In this region, surface $pCO_2$ observations show a marked $pCO_2$ decline near the Mississippi delta during spring-summer (see Fig. S6 in the Supplement). This pattern is not well reproduced by Xue et al. model, as their model overestimated $pCO_2$ on the inner northern GoM shelf. In addition, Xue et al. underestimated surface $pCO_2$ in the open GoM, obtaining a marked $pCO_2$ minimum over the Loop Current region (see their Fig. 13a), the latter a feature that is not supported by observations (Fig. S6). Consequently, differences between the $CO_2$ fluxes derived from our model and those derived from Xue et al. can be mainly explained by $pCO_2$ biases in Xue et al. model. We will include a brief discussion in Section 6.2 on the differences between our results and those reported by Xue et al (2016).*

*Patterns of high DIC removal and nutrient depletion have been documented for the MARS delta region (e.g. Guo et al., 2012) and are also reproduced by our model. Note that the net community production patterns shown in Fig. 7 represent the balance between primary production (DIC uptake) and respiration, and the simulated primary production can be straightforwardly translated to nitrogen uptake using Redfield ratios.*

**Technical Corrections**

*Technical corrections indicated by the second reviewer will be incorporated in the new manuscript version. Below we show specific answers for few of them.*

**L98,99: I do not see a description of how to calculate river DIC from river (pH, TA, T) in the Stets et al. (2014) listed in the References. Is this the correct reference?**

*We thank the reviewer for noting this mistake. The correct reference is:*

*Stets, E. G., and Striegl, R. G.: Carbon export by rivers draining the conterminous United States. Inland Waters, 2(4), 177-184, 2012.*

**Figure 2: Assuming this is practical (not absolute) salinity, I disagree that it should be "unitless". You could neatly specify which definition is used with the unit [psu], which for me is a perfectly valid and informative dimensionless unit.**

*We prefer keep salinity unitless (See Millero, F.J. 1993. What is PSU? Oceanography6(3):67)*

**Figure 3: Add sentence to caption saying what the red/blue lines are (presumably mean values over all model grid points and observations within the regions defined in Fig. 1). Reference should be to Fig. S5 not S4.1. Refer to northern GoM SHELF (assuming that the statistics are restricted to the shelf region).**

*The legend for Figure 3 will be modified to:*

*Mean monthly patterns for the observed (red lines) and simulated (blue lines) surface $pCO_2$ over the (a) open GoM and (b) northern GoM regions (shown in Fig. 1). Light pink and cyan shades depict the observed and modeled interquartile interval, respectively. Gray shades depict the model's 5%-95% percentile interval. Observations are from Ships of Opportunity and Research Cruises conducted during 2005-2014 (ship tracks are shown in Fig. S4.1).*

**References**

*Gomez, F. A., Lee, S.-K., Liu, Y., Hernandez Jr., F. J., Muller-Karger, F. E., and Lamkin, J. T.: Seasonal patterns in phytoplankton biomass across the northern and deep Gulf of Mexico: a numerical model study, Biogeosciences, 15, 3561-3576, https://doi.org/10.5194/bg-15-3561-2018, 2018.*

*Guo, X., et al.: Carbon dynamics and community production in the Mississippi River plume, Limnology and Oceanography 57.1, 1-17, 2012.*

*Huang, W. J., Cai, W. J., Wang, Y., Lohrenz, S. E., and Murrell, M. C.: The carbon dioxide system on the Mississippi River dominated continental shelf in the northern Gulf of Mexico: 1. Distribution and air-sea CO2 flux, J. Geophys. Res.-Oceans, 440(120), 1429–1445, 2015.*

*Liu, Y., Lee, S. K., Enfield, D. B., Muhling, B. A., Lamkin, J. T., Muller-Karger, F. E., and Roffer, M. A.: Potential impact of climate change on the Intra-Americas Sea: Part-1, A dynamic downscaling of the CMIP5 model projections, J. Mar. Syst., 148, 56–69, 2015.*

*Xu, Y.-Y., W.-J. Cai, Y. Gao, R. Wanninkhof, J. Salisbury, B. Chen, J. J. Reimer, S. Gonski, and N. Hussain: Short-term variability of aragonite saturation state in the central Mid-Atlantic Bight, J. Geophys. Res. Oceans, 122, 4274–4290, doi:10.1002/2017JC012901, 2017.*

*Xue, Z., He, R., Fennel, K., Cai, W.-J., Lohrenz, S., Huang, W.-J., Tian, H., Ren, W., and Zang, Z.: Modeling pCO2 variability in the Gulf of Mexico, Biogeosciences, 13, 4359–4377, 2016.*

---

## Author Response (AR1)

**Response to the Reviewer Comments**

We highly appreciate the reviewers for all their valuable comments and suggestions. In this document we provide the responses to the reviewers comments, as well as the new manuscript version with the 'track change' option.

**Response Reviewer 1:**

**My main general comment is to expand the discussion of your results. Currently it is not clear what is the novelty of your estimates in air-sea CO2 flux in comparison to the previous regional work. Your discussion mostly focuses on showing that your model is able to simulate the regional carbon dynamics. Could you also provide some discussion what your results mean in term of the Gulf regional carbon dynamics, i.e. for each of your regions? For instance you mention that there is a need "to identify coastal ecosystem susceptibility to ocean acidification" but this is not discussed further, i.e. with respect to your results.**

*The following sentences were added to the new manuscript version to expand the discussion on sea-air CO$_2$ fluxes:*

*Lines 336-341:*
*"Finally, the simulated annual carbon uptake was weak for most of the GoM basin. Therefore, it is likely that relatively small disturbances in the pCO$_2$ drivers could turn the carbon sink regions into carbon sources. A potential mechanism for this change is ocean warming, since future ocean projections in the GoM suggest a significant SST increase (>2°C) due to anthropogenic climate change to the end of the twenty-first century (Liu et al., 2012; 2015; Alexander et al., 2020; Shin and Alexander, 2020). This is a topic deserving examination for future modeling efforts."*

*Lines 371-378:*
*"The simulated fluxes largely differ from the fluxes reported by Xue et al. (2016), which was the only*

*previous regional modeling study describing basin wide patterns in the GoM. They obtained a three times stronger uptake in the open GoM, and much weaker uptake on the shelf regions (e.g. their simulated annual flux for the northern GoM shelf was one third of our estimation). We believe these differences in $CO_2$ fluxes can be mainly explained by $pCO_2$ biases in the model used in Xue et al. (2016). Indeed, their model underestimated surface $pCO_2$ in the open GoM, and thus obtained a marked $pCO_2$ minimum over the Loop Current region (see their Fig. 13a), a pattern not supported by SOOP observations (Fig. S6). In addition, their model largely overestimated surface $pCO_2$ on the northern GoM and west Florida inner shelves, especially during summer-fall, not reproducing well the marked $pCO_2$ drop that is observed close to the MARS delta."*

*We also included the following paragraph to link the derived patterns of surface $\Omega_{Ar}$ with ecosystem susceptibility to ocean acidification:*

*Lines 304-314:*
*"Surface $\Omega_{Ar}$ patterns can be useful to identify regions more vulnerable to ecosystem disturbances induced by surface ocean acidification. Our model indicates minimum surface $\Omega_{Ar}$ ranging from 2.5 to 3.4 on the northern GoM and west Florida inner shelves during winter, and greater than 3.4 on the western GoM and Yucatan shelves. This suggests higher ecosystem resilience to surface ocean acidification in the latter regions. Surface $\Omega_{Ar}$ patterns do not necessarily reflect vulnerability of coastal benthic organisms to ocean acidification, since $\Omega_{Ar}$ values for surface and bottom layers can largely differ in regions where the water column is strongly stratified. This is the case for the Louisiana inner shelf during summer, which displayed maximum surface $\Omega_{Ar}$ values (>4.2) linked to high biological uptake, but low bottom $\Omega_{Ar}$ values (<2.6; not shown) due to bottom acidification induced by organic carbon remineralization and weak bottom ventilation (see Cai et al. (2011) and Laurent et al. (2017) for further discussion). However, our model outputs did not reveal such signature of bottom acidification on the west Florida, western GoM and Yucatan shelves, as these regions display relatively weak vertical stratification and lower eutrophication levels compared to the northern GoM shelf."*

**Specific comments:**

**L106-107: is it 40/9 years of the same annual cycle? Is 9 years sufficient for the carbon system, e.g. for the deep Gulf it seems short, and how did you assess that the carbon system was spun-up?**

*We consider that a 9-year simulation is an appropriate time for spinning-up the model, as the simulated DIC and TA patterns in the upper ~800 m reached a periodic steady state after 3 or 4 years. This was checked by visual inspection of the model outputs, and estimating DIC and TA linear trends. Deep layer pattern could take longer, but this most likely has a limited influence in the surface properties of the GoM.*

**L109: It would have been interesting to see model output for the period January 1981 to November 2014.**

*We focused on seasonal dynamics only in this paper because interannual variability will be addressed in a following study.*

**L144-145: a more accurate statement would be "Overall, simulated and observed pCO2 patterns agreed with observations"**

*The change was done accordingly.*

**L146-147: is it possible that your model generally overestimates surface primary production, resulting in a lower surface pCO2?**

*Although an overestimation of surface primary production could not be discarded, the comparison between simulated and observed primary production pattern did not reveal any evident bias (see Fig. 6 in Gomez et al., 2018).*

**L144-151: There is obviously a very large difference in the shape of the observed versus modeled pCO2 time series in NGoM (Figure 3b). This should be discussed. Why is there a strong dip in pCO2 in March in the observations and why it doesn't occur in the model? January-February observations are odd. Figure 3b also shows that there is a 1 month delay in the modeled pCO2, which tend to follow more the temperature cycle. Can you discuss these discrepancies?**

*The pCO2GoM_2018 dataset has very few observations during wintertime. Only 8% of the GoM data were collected in December-February, less than 2% during January. Indeed, for the northern GoM, the observations for January were derived from only one cruise. Consequently, there is a large observational uncertainty during winter. We included the following sentences to clarify:*

*"In the northern GoM, the largest disagreement was observed in January-February (Fig. 3b), but this difference is most likely due to the reduced number of observations during winter in the pCO$_2$GoM_2018 dataset (Fig. S6). Indeed, January observations came from only one cruise, which largely increases observational uncertainty."*

**L152-160: in Figure 4 caption can you add location information, i.e. off Tampa (upper panel?) and off Louisiana (lower panel?) and refer to Figure 4a and Figure 4b when appropriate.**

*The change was done accordingly.*

**L155-156: the 0-200m difference in DIC and TA is quite large. You need to provide more discussion here to gain confidence in the results presented below. What is the source of this discrepancy?**

*We recognize that there are important differences between modeled and observed profiles in the upper 200 m for the Mississippi line, although observed values are within or close to the simulated variable's*

*range. The open GoM region off Louisiana, where the Mississippi line extends, is strongly influenced by the Mississippi river runoff, displaying a large spatiotemporal variability. Relatively minor differences between the observed and simulated cross-shore fluxes could lead to important differences between the observed and simulated vertical distribution of DIC and TA. We included the following sentences in the new manuscript version:*

*"Monthly averaged model DIC and TA were underestimated in the upper 200 m off Louisiana, with the bias ranging from around 5 to 90 µmol kg$^{-1}$ for DIC and 5 to 40 µmol kg$^{-1}$ for TA, but the observations were within or close to the simulated variable's ranges during June-August 2000-2014. These model-observation differences could be partly due to misrepresentation of cross-shore transport in a region strongly influenced by the Mississippi river runoff."*

**Response Reviewer 2:**

This is a very well written manuscript, with good figures, and scientific arguments that are interesting and well-constructed. The authors appear to have made a significant advance in understanding the regional and temporal variability of seawater CO2 chemistry and air-sea CO2 fluxes in the Gulf of Mexico. There is potential for expanding the discussion (see specific comments). However, one of the aspects I appreciated of the manuscript as it stands was the concise length: the authors should try to limit text additions in response to reviewers and balance with (careful) trimming. Overall, I congratulate the authors for an excellent submission and recommend only minor revisions.

Specific Comments

"air-sea flux": Most readers will read this as AIR-TO-SEA flux. Therefore it should be positive when the ocean is a sink and negative when the ocean is a source (as in e.g. Xue et al., 2016). I recommend that the authors either: a) reverse the signs on all "air-sea" fluxes, or b) speak of

**"sea-air" fluxes, which is an alternative convention.**

*This new paper version uses "sea-air" flux instead of "air-sea".*

**L112-113: The carbonate chemistry equilibrium constants from Mehrbach et al. (1973) refitted by Dickson and Millero (1987) might not be optimal for the salinities <20 psu explored by this study (cf. Fig. 6; see Millero 2010, and stated validity ranges in the van Heuven code).**

*Although using Millero (2010) constant for low salinity regions (S<20) could be optimal, the deviation between Millero and Mehrbach et al. is not significant until S<5. Very little of the study area has S < 20. Indeed, due to the spatial resolution of our model (~8 km), the simulated surface salinity values were always greater than 5, and only a small fraction (0.1%) of the surface layer outputs have salinities smaller than 20. Therefore, we believe that using Mehrbach et al. constants does not significantly bias our model results.*

**L148-150: Looking at Fig. S6 it seems that the mismatch may be primarily due to interannual variability and temporal undersampling. The observations in Fig. S6 do suggest a strong decrease in pCO2 on the northern GoM shelf during JFM, but they appear to be from single cruises and perhaps a single year (?), while the model results are averaged over 10 years. Perhaps clarify about this.**

*In the northern GoM, the observations for January, February, and March were derived from one (2009), three (2009, 2010, 2012), and two (2009, 2010) cruises, respectively. Certainly, the limited number of pCO₂ observations collected during wintertime increases the observational uncertainty. We included the following sentences to clarify:*

*"In the northern GoM, the largest disagreement was observed in January-February (Fig. 3b), but this difference is most likely due to the reduced number of observations during winter in the*

*pCO₂GoM_2018 dataset (Fig. S6). Indeed, January observations came from only one cruise, which largely increases observational uncertainty."*

**L272-273: I am left wondering exactly why, compared with the Xue et al. model, the present model apparently simulates stronger biological DIC uptake and associated pCO2 decrease in the MARS region, sufficient to turn this region into a year-round CO2 sink (cf. present Fig. 11 vs. Xue et al., 2016, Fig. 7). Is it possible that the apparent improvement in fit to surface pCO2 observations (present Fig. S6) could be for the wrong reasons? Can the stronger biological uptake be corroborated with other observations (e.g. nutrient drawdown)? Also, it seems that the large-scale seasonal variability in CO2 flux, here driven primarily by temperature, is stronger than in Xue et al. Is this true, and if so, why?**

*We feel confident that our model produces realistic results. A series of studies have recognized the importance of biological uptake as a main driver of carbon patterns in the northern GoM (e.g. Guo et al. (2012) and Huang et al. (2015)). In this region, surface pCO₂ observations show a marked pCO₂ decline near the Mississippi delta during spring-summer. This pattern is not well reproduced by Xue et al. model, as their model overestimated pCO₂ on the inner northern GoM shelf. In addition, Xue et al. underestimated surface pCO₂ in the open GoM, obtaining a marked pCO₂ minimum over the Loop Current region. Consequently, differences between the CO₂ fluxes derived from our model and those derived from Xue et al. can be mainly explained by pCO₂ biases in Xue et al. model. We have added the following paragraph into the Discussion section:*

*"The simulated fluxes largely differ from the fluxes reported by Xue et al. (2016), which was the only previous regional modeling study describing basin wide patterns in the GoM. They obtained a three times stronger uptake in the open GoM, and much weaker uptake on the shelf regions (e.g. their simulated annual flux for the northern GoM shelf was one third of our estimation). We believe these differences in CO₂ fluxes can be mainly explained by pCO₂ biases in the model used in Xue et al. (2016). Indeed, their model underestimated surface pCO₂ in the open GoM, and thus obtained a marked*

*pCO₂ minimum over the Loop Current region (see their Fig. 13a), a pattern not supported by SOOP observations (Fig. S6). In addition, their model largely overestimated surface pCO₂ on the northern GoM and west Florida inner shelves, especially during summer-fall, not reproducing well the marked pCO₂ drop that is observed close to the MARS delta."*

**Technical Corrections**

*Technical corrections indicated by the second reviewer were incorporated in the new manuscript version. Below we show specific answers for few of them.*

**L98,99: I do not see a description of how to calculate river DIC from river (pH, TA, T) in the Stets et al. (2014) listed in the References. Is this the correct reference?**

*We thank the reviewer for noting this mistake. The correct reference is:*

*Stets, E. G., and Striegl, R. G.: Carbon export by rivers draining the conterminous United States. Inland Waters, 2(4), 177-184, 2012.*

**Figure 2: Assuming this is practical (not absolute) salinity, I disagree that it should be "unitless". You could neatly specify which definition is used with the unit [psu], which for me is a perfectly valid and informative dimensionless unit.**

*We preferred keeping salinity unitless (See Millero, F.J. 1993. What is PSU? Oceanography6(3):67)*

**Figure 3: Add sentence to caption saying what the red/blue lines are (presumably mean values over all model grid points and observations within the regions defined in Fig. 1). Reference should be to Fig. S5 not S4.1. Refer to northern GoM SHELF (assuming that the statistics are restricted to the shelf region).**

*The legend for Figure 3 was modified to:*

*"Mean monthly patterns for the observed (red lines) and simulated (blue lines) surface pCO₂ over the (a) open GoM and (b) northern GoM regions (shown in Fig. 1). Light pink and cyan shades depict the observed and modelled interquartile interval, respectively. Gray shades depict the model's 5%-95% percentile interval. Observations are from Ships of Opportunity and Research Cruises conducted during 2005-2014 (ship tracks are shown in Fig. S4.1)."*

[revised manuscript text omitted]